# Interfacial Behaviour in Polymer Composites Processed Using Droplet-Based Additive Manufacturing

**DOI:** 10.3390/polym14051013

**Published:** 2022-03-03

**Authors:** Sofiane Guessasma, Khaoula Abouzaid, Sofiane Belhabib, David Bassir, Hedi Nouri

**Affiliations:** 1INRAE, UR1268 Biopolymères Interactions Assemblages, Rue de la Géraudière, F-44300 Nantes, France; khaoula.abouzaid@inrae.fr; 2Oniris, CNRS, GEPEA, UMR 6144, Université de Nantes, F-44000 Nantes, France; sofiane.belhabib@univ-nantes.fr; 3LMC, UMR-CNRS 5060, Université Bourgogne Franche-Comté (UTBM), F-90010 Belfort, France; david.bassir@urbm.fr; 4Centre Borelli, ENS-University of Paris-Saclay, F-91190 Gif-sur-Yvette, France; 5Laboratory of Electromechanical Systems (LASEM), National Engineering School of Sfax, University of Sfax, Sfax 3038, Tunisia; hedi.nouri@isams.usf.fr

**Keywords:** droplet-based additive manufacturing, X-ray micro-tomography, finite element computation, thermoplastic polyurethane elastomer

## Abstract

In this study, we show the extent of interfacial behaviour in the mechanical performance of thermoplastic polyurethane elastomer (TPU)/acrylonitrile butadiene styrene (ABS) composite material manufactured using droplet-based additive manufacturing. Both the interface orientation and the interface strength are varied during the processing. Prior to tensile experiments, X-ray micro-tomography imaging is undertaken to obtain the microstructural arrangement of polymer droplets in the part. Tensile loading is performed simultaneously with digital image acquisition to reveal the extent of strain localization using a digital image correlation approach. The experiments are performed up to the failure of the specimens. Finite element computation based on 3D imaging of the ABS/TPU composite is considered to predict the role of the interface as well as the defect influence on the tensile performance. The experimental results show a major connectivity of the process-generated porosity and a distinct morphology of the ABS/TPU interface. The predictions demonstrate that, despite the limited amount of porosity, their connectivity plays a significant role in triggering damage initiation and growth up to the failure of the composite material.

## 1. Introduction

The manufacturing of technical parts from a digital model is an effective way of producing complex geometries [1,2]. As reported by [1], AM is developed to conceptualise models for products but quickly to process materials. Zai et al. [2] also reported that AM is an advanced manufacturing process for the fabrication of 3D functional components with a limited dependence on tooling and fixture. Despite the large amount of time needed to manufacture a single part by additive manufacturing (AM) technologies compared to traditional processes such as injection moulding or extrusion [3], AM is still an attractive method [3,4,5]. Indeed, the fabrication cycle for AM requires only a CAD model instead of preparing, adapting, or producing a new tool. As described by Baumers et al. [3], AM has, in that sense, a huge economic impact on the manufacturing sector because of the reduced tooling costs. According to Berman [5], this reduction in cost can be also attributed to automation, which allows AM be autonomous, not requiring constant attention from the operators during the production cycle. The economic impact has been covering a large number of industries, but as reported in the review paper by Mousa et al. [4], key sectors such as aerospace, automotive, medical industries, and electronics are sharing most of the AM focus. In terms of the fabrication cycle, once the CAD model is prepared, a series of tool paths is the only intermediate step to transform the virtual sketch into a real part [6,7,8]. As explained in the paper by Pandey [6], the series of tool paths requires slicing the CAD model into a series of thin layers. The thickness of these layers determines the balance between the surface finish and the building time. According to Hague et al. [7], the outcome of the slicing procedure results in critical factors such as surface finish, resolution, and accuracy that engineers need to constantly adapt to allow AM to be a viable manufacturing technology. This is why early contributions such as the one on stereolithography by Jamison et al. [8] tackled the deviation errors in design tessellation, which is the basic procedure for converting the 3D design into a set of triangular patches.

Waiving the obligation for tooling means also increasing the offer for personalised products that AM is able to fulfil [5]. There is no doubt that several industries clearly understood this potential by directing a substantial investment towards AM-based products [9,10,11,12,13]. The review paper by Sing et al. [9] shows that the leading market for AM-based products is consumer products and electronics, with up to 21% of the total market shares. The automatic and medical sectors come in the second and third positions, respectively. In the automotive industry, for instance, the market needs lighter, safer, and cost-effective products with as short a lead time as possible, which are the main value drivers for AM according to Giffi et al. [10]. In the aerospace sector, safety concerns with regards to the reliability of 3D printed parts is still a challenge, but some leading projects using AM processes to manufacture parts have been granted substantial funding, such as to produce air-cooling ducts, plastic interior parts, leading-edge blades, and even entire spacecraft structures [11]. In the medical sector, Peltola et al. [12] pointed out the advantage of the overall conventional AM route for biocompatible parts for customising implants or engineered body parts from medical scans of patients. A huge segment of the market is dedicated to AM technologies, such as tissue engineering for bone and cartilage repair applications. For civil engineering applications, Krimi et al. [13] demonstrated that design flexibility versus the production rate is still a challenging balance for AM to compete with standard routes in construction such as prefabrication or casting on site. The growing interest in AM technologies comes with some aspirations to solve the problems of part inaccuracy and finishing state [14,15,16], to improve the productivity [9], and to gain in performance [17,18]. For instance, Zhou et al. [14] applied surface response and ANOVA analysis to optimise the process parameters in stereolithography technology to achieve optimal dimensional accuracy and build time. Most of the performance limitations come from mechanical anisotropy and the loss in strength and stiffness driven by the presence of the particular arrangement of defects. For instance, Dawoud et al. [18] compared the rendering of FDM (fused deposition modelling) with injection moulding and showed that ABS specimens manufactured using injection moulding exhibited the highest mechanical performance. Hague et al. [7] compared the mechanical performance of specimens printed using stereolithography and the laser sintering process under different building orientations. The study concluded on different trends with respect to the processing route. Although stereolithography based specimens exhibited isotropic properties, the laser sintering process generated more anisotropic parts. Puebla et al. [17] showed, however, that stereolithography may generate anisotropic behaviour, especially if the part orientation is combined with different aging trends. Huang et al. [15] studied the shape shrinkage responsible for design dimension inaccuracy and developed a methodology for anticipating geometric errors. Depending on the complexity of the designs, geometrical inaccuracy can also be related to the presence of support material needed for printing overhangs. Thus, attempts have been made to decrease the influence of support material as suggested by Paul et al. [16] by using an algorithmic approach to minimise the flatness and cylindricity errors.

There is, generally, no balance between AM technologies with regards to their sensitivity to process-induced defects. For instance, fused deposition modelling (FDM) is known to produce two-dimensional discontinuities because the process is based on the laying down of 1D continuous features, i.e., the filaments. In 3D space, this obviously generates a particular arrangement of three-dimensional porosities. As a result, the part contains a well-ordered porosity network evidenced by several imaging techniques such as X-ray micro-tomography [19]. The origin of mechanical anisotropy, and especially the lack of performance in the building direction, results from this process-induced porosity. In the case of stereolithography, the laser drawing leaves no material discontinuity within the plane, and the liquid state of the resin used in such technology ensures the continuity of the material both in the plane of deposition and along the building direction [20]. This means that stereolithography has better mechanical rendering compared to FDM but has some limitations, especially in producing parts with closed porosities [20]. In addition, stereolithography requires the feedstock material to be photo-sensitive and generates more operating costs. These factors narrow down the spectrum of applications for this type of technology.

In this study, the effect of process-induced defects in droplet-based additive manufacturing is investigated with respect to the tensile performance of a polymeric composite. This technology naturally produces 3D discontinuities related to the successive laying down of individual droplets. In a printed composite structure, the material discontinuity may exhibit a complex trend that this study intends to quantify. For instance, in a two-phase composite, defects created in each phase are not the same as those dependent on the thermal behaviour of the intrinsic phases during the laying down. However, close to the interface, both phases are expected to contribute to the defects created by material discontinuity. In particular, this study combines a 3D imaging tool based on X-ray micro-tomography, finite element computation, and testing experiments to reveal the nature, magnitude, and extent of the process-generated defects as well as the interfacial effect on the mechanical response. The materials considered for this study are acrylonitrile butadiene styrene (ABS) and thermoplastic polyurethane (TPU). ABS material is usually used for applications where chock resistance, dimensional stability, and electric insulation are required. It is, for instance, widely used in consumer products, communication equipment, automotive parts, and home appliances. TPU, on the other hand, is known for its elasticity and resistance to abrasion, and has common use as ABS in the automotive, appliance, construction, and recreation industries. The combination of the tough and elastic properties in a single material is needed in some applications like in the automotive industry to allow damping under cycle loading while maintaining a sufficient mechanical resistance. Thus, this study focuses on understanding how the composite made of ABS and TPU can be used without degrading the performance of the two phases. This can be done by looking at the interface bounding when using bi-material parts in the context of a droplet 3D printing technique. Indeed, 3D printing technologies allow more degrees of freedom to adapt the quality of the interface as in the droplet-based technology to achieve adaptable designs with the ability to combine different materials.

In fact, the overmoulding process can be considered the conventional counterpart of AM for the design of composite structures [21]. Although substantial progress has been made in the area of multi-material injection moulding, the complexity of the designs brought on by the growing need for light-weight and high-performance composites limits the use of the overmoulding process to simplified designs. Another critical issue addressed in many papers related to the overmoulding technique is the quality of the interface [22], where in these conventional processes, there is a limited leverage to adjust the interface properties by directly playing on the process deposition. This context allows the droplet-based technology to be a step ahead in providing leverage both at the material side by selecting the appropriate material combination and at the process side by locally adjusting the interface properties.

This approach is considered in this study to demonstrate the leverage on the design that can help improve the bonding between ABS and TPU. Composites exhibit strong or weak interface properties depending on the nature of chemical bond generated at the interface. In the present study, both the thickness and orientation of this interface are varied to quantify the quality of ABS/TPU interface properties on the composite tensile behaviour.

## 2. Experimental Layout

The two polymeric materials ABS and TPU were used to manufacture regular samples with the dimensions 70 mm × 10 mm × 4 mm. In the three-dimensional space attached to the CAD model of the sample, the following conventions were used for each axis (Figure 1): X—width direction, Y—longitudinal direction, and Z—thickness or building direction.

The in-plane (X–Y) surface denotes the plane normal to the building direction and containing the two largest dimensions of the sample (length and width). The two materials were provided by BASF GmbH, Berlin, Germany, under the trade names of Terluran GP35 Green and Elastollan EC78A15. The two polymers had similar densities (1.18 g/cm^3^ for TPU and 1.04 g/cm^3^ for ABS) and close values of Poisson ratios (0.419 for TPU and 0.394 for ABS), but a large difference in Young’s moduli (527 MPa for TPU and 2300 MPa for ABS).

3D printing was performed using the Arburg Plastics Freeformer from Arburg, Germany (Figure 1b). Droplet deposition was performed at a rate of 80 mm/s and with a 100% infill (duration per print was 35 min.). The slice thickness was adjusted to 0.21 mm, assuming a shape factor for the droplet of 1.04. Two extrusion heads of 150 µm in diameter each were used to print the ABS/TPU composite material following the logic explained in Figure 1. The interface orientation with respect to the loading direction is labelled by the angle θ. Both horizontal (θ = 90°) and inclined (θ = 60°) interfaces were considered in the study. The number of droplets (n) that spilled over the interface was adjusted to one or two intertwining droplets. As suggested in Figure 1, the laying down of the droplets was performed along the thickness of the composite sample. There were four different situations, named AT01 through AT04, that were tested to reveal the interfacial behaviour of the ABS/TPU composite. The interphase covered a finite thickness of 0.21 mm or 0.42 mm, depending on the printing conditions. The droplet arrangement in Figure 1 was secured by an external frame to provide more strength to the sample.

X-ray micro-tomography was used to obtain the 3D arrangement of the droplets after printing. The imaging setup based on UltraTom X-ray µ-CT from Rx-Solutions (Chavanod, France) equipment and sample views are shown in Figure 2.

The central part of the ABS/TPU specimens including the interface zone was imaged with typical ROI volumes in the range of 10 mm × 10 mm × 4 mm and 14 mm × 11 mm × 5 mm. The imaging conditions were as follows: voltage = 60 kV, current intensity = 70 µA, resolution = 1920 × 1536 pixels, and voxel size between 8.36 µm and 10.92 µm.

The construction of the 3D tomograms based on 1440 radiographic 2D images per sample was performed using the filtered back-projection technique using the X-Act software environment from Rx-Solutions (Chavanod, France). The acquisition time was about 23 min per sample. The number of voxels per tomogram was in the interval of 0.55–0.84 billon voxels.

Image analysis was considered to process the 3D images and isolate all features of interest, including the ABS and TPU phases, the interface region, and the process-induced porosity. ImageJ free software from NIH (version 1.8.0_172, Bethesda, MD, USA) was used with typical operators such as 2D/3D labelling, dilation, erosion, particle analysis, thresholding, flooding, wrapping, and image calculators. Porosity content was derived from segmented images, where the porosity phase appeared in binary images as black voxels ($g_{i}=0$) and white voxels ($g_{i}=255$) were assigned to either the ABS or TPU phase. Further processing treatment was done to avoid counting the surrounding external area and to distinguish between the porosity levels in each phase.
(1)$$f\left(\%\right)=\sum_{j}^{i\in \prod }i\left(\left(255-g_{i}\right)/255\right)/\sum_{j}^{i\in \prod }g_{j}$$
where *g_i_* is the grey level of voxel *i* from the domain ∏, where the porosity level was measured.

Tensile experiments were performed on Lloyd universal testing equipment from Lloyd Instruments Ltd. (Sussex UK), (maximum load capacity of 10 kN). The displacement rate was fixed to 5 mm/min for a gage length in the range of 41–42 mm. From the tensile experiments, both engineering strain and stress were derived (Figure 3c) and the Young’s modulus was measured from the initial slope of the stress–strain curves.

Tensile loading was simultaneously conducted with digital image correlation instrumentation using equipment provided by Correlated Solutions (Irmo, SC, USA). Optical recording was performed with a resolution of 2000 × 2000 pixels (where the size of one pixel was 166 µm) and constant illumination conditions to allow the exploitation of the produced sequences of grey level images with the VIC-2D software. The acquisition frequency was adjusted to 5 frames per second (fps) and the subset size was limited to 2.5 mm and stepped by 0.8 mm. The accuracy of the measurement was as small as 10 microstrains, which is well below the strain levels achieved upon loading. Figure 3 shows the experimental setup and ruptured samples covered with a layer of speckles needed for the image correlation process.

## 3. Modelling Technique

Two main models are explored in this section. The first one is a macroscopic model based on the geometry of the printed sample. The meshing of the paralepipedic sample was preformed using a reasonable number of 3D-8node structural elements (13,400 elements). Each node is capable of translations in the X-, Y-, and Z-directions. In addition, interface elements were added as 3D-8node cohesive zone elements. The model size was thus increased to 48,762 degrees of freedom (dof). Both ABS and TPU phases were considered as isotropic elastic materials, fully defined by the Young’s moduli and Poisson ratios introduced earlier. In addition, a cohesive model was implemented [23]. According to this model, the work $W_{I}$ at the interface can be written as
(2)$$W_{I}=\int T\times \delta \times ds$$
where *T* is the traction at the interface, δ is the interfacial separation, and *ds* is a surface normal to the interface.

The interfacial separation and the traction can be expressed as a function of the normal and tangential components as follows:(3)$$d_{n}=\delta \times n;\;d_{t}=\delta \times t;\;T_{n}=T\times n;\;T_{t}=T\times t;\;$$
where $d_{n}$ and $d_{t}$ are the normal and tangential separations, respectively; $T_{n}$ and $T_{t}$ are the normal and tangential components of the traction, respectively; and n and t are unit vectors in tangential and normal directions, respectively.

An exponential function of the cohesive zone model was adopted according to the surface potential *φ* as follows:(4)$$\varphi \left(\delta \right)=2.71\times \sigma_{I}\times \delta_{n}\times \left(1-\left(d_{n}/\delta_{n}\right)\times e^{-\left(d_{n}/\delta_{n}\right)}\times e^{-{\left(d_{t}/\delta_{t}\right)}^{2}}\right)$$
where $\delta_{n}$ and $\delta_{t}$ are the maximum normal and tangential separations, respectively, and $\sigma_{I}$ is the maximum normal traction. These were the three interfacial parameters to be identified based on the experiments.

Image-based finite element (FE) computation was considered to predict the tensile performance of the ABS/TPU printed samples. The grey-level information contained in the tomograms was converted into a material model. The computational domain corresponded to the central part of the specimen imaged with the dimensions given above. The computational model was thus based on a voxel-to-cubic-element conversion under the simulation environment Ansys Mechanical (Canonsburg, PA, USA). Cuboid elements defined by eight nodes and with three degrees of freedom each (structural displacements in the X-, Y-, and Z-directions) were used to mesh the printed microstructure. The meshing was conducted on a regular basis, which means that the model size was linearly proportional to the resolution of the embedded 3D images. Preliminary tests conducted on a high-performance computer (dual processor E-5 3Ghz equipped with 1Tbytes of RAM) showed that decreasing the original resolution of the tomograms by a factor of between 6.67 and 8.42 was required to achieve the largest possible models on the available resources. This limit corresponded to a maximum model size of between 153 and 257 million degrees of freedom, depending on the resolution of the tomograms. The element size varied between 18 µm and 22 µm. In addition, nine other intermediate meshes were used to check the sensitivity of the computations with regards to the resolution lowering. The largest element size varied for these cases between 145 µm and 218 µm, depending on the printing conditions. Four different grey levels were associated with four different phases, namely, ABS, TPU, interphase, and porosity. The elasticity constants for the first two phases were the ones introduced in the experimental section. Material properties for the interphase layer were identified from the experiments. The first load increment was used to identify the Young’s modulus of the composite for all conditions. While the loading was increased, a damage model was implemented based on a stress intensity criterion. For the elements that reached the maximum stress intensity, their associated Young’s modulus was changed to the ground value. The load was increased up to the rupture point corresponding to a residual reaction force. The number of damaged elements with respect to the total number of elements that could be possibly damaged scaled the damage variable. The damage kinetics were plotted by monitoring the damage variable as a function of the iteration level, which refers to the number of load levels.

For both macroscopic and microscopic models, displacement constrain conditions were used to simulate the tensile loading as follows:(*U_x_* = 0, *U_y_* = 0, *U_z_* = 0); *z* = 0(5)
and
(6)$$\left(U_{x}=0,U_{y}=0,U_{z}=\Delta U\left(t\right)>0\right)\;;\;z=L$$

Where $U_{x}$, $U_{y}$, and $U_{z}$ are translations or structural displacements in the X-, Y-, and Z-directions, respectively; L is the sample length or the largest dimension of the microstructure; ΔU is a positive amount, and *t* is the iteration level.

The elasticity problems were solved using a preconditioned conjugate gradient solver with a typical computation duration of less than 5 min for the macroscopic model and between 229 and 346 min for the largest models based on X-ray micro-tomography images. The PCG error tolerance was adjusted to 10^−8^, which is the recommended value for elasticity problems.

## 4. Results and Discussion

### 4.1. Microstructural Characteristics

Figure 4a shows the relative contrast between the phases achieved from the as-acquired X-ray images. The radiographic image of sample AT03 is a configuration represented by an inclined interface (q = 60°) and one intertwining droplet at the interface (n = 1). The meaningful variation in grey levels corresponded to the interphase layer and the surrounding air. Figure 4b shows an in-plane cross-section view (X–Y plane) including the longitudinal (Y) and width (X) directions. This in-plane view corresponded to one of the slices of a constructed tomogram from the series of 2D radiographic images of sample AT03. A significant connectivity between micro-scaled porosities is the main highlighted feature. This porosity was generated because of the space left behind the successive lines of droplets during the additive manufacturing process. A second type of porosity was also generated between the external frame and droplet raster due to the abrupt change in printing nozzle trajectory. The third type of porosity was generated at the interface, and this one marked the lack of adhesion between the ABS and TPU phases. In addition to these types of porosities, necking along the linear arrangements of droplets was evidenced despite the overlap parameter selected for printing. It is worth mentioning that the relative contrast between ABS and TPU was too small to allow segmentation of the image based on a simple grey-level threshold. Figure 5 illustrates the rendering of a segmentation procedure involving flooding techniques and 2D and 3D labelling of features of interest. To ease the identification of all features of interest such as the porosity, some of the features were deliberately removed along a few slices from the thickness.

Figure 5 shows the particular pore connectivity generated by the droplet-based AM. Indeed, the porous network shared the same characteristics as the droplet raster, with crossing angles at +45°/−45° arranged perpendicularly along the sample thickness (building or Z-direction). The isolation of the external air allowed for the evaluation of the surface finish state in Figure 5, which clearly demonstrates the presence of a regular grid of small pits.

Quantitative evaluation of the features of interest, namely, porosity and the ABS and TPU phases, was performed based on the results of the segmentation. Figure 6 depicts the porosity content ($p_{i}$) profiles in all space directions. These profiles were obtained by summing up all the pixels with the grey level assigned to the porosity feature ($g_{jk}$) over the surface area ($A_{i}$) evaluated at each slice i:(7)$$p_{i}\left(\%\right)=100\times \sum_{j,k}^{}g_{jk}/A_{i}$$
where j,k are the indices running through the image slice.

Figure 6a shows the depth profile of the porosity arrangement in the Z-direction (building direction) for all studied configurations.

The achieved sinusoidal-like variation of porosity content suggests a regular arrangement of process-generated porosities throughout the thickness of the specimens. There was no marked anisotropy nor a sharp gradient measured in this direction. The lowest magnitude of variation in porosity content corresponded to the case AT03. The largest peaks in porosity content reached 10% for AT04. It was not possible to obtain the distinct effect of the interface on the generated porosity from the reading of the depth profiles because all the depth slices contained the interface layer (Figure 7). The cross-section views suggest, however, that the presence of the frame at the interface added more complexity in the toolpath trajectory, which in turn generated a certain additional amount of porosity at the interface. The cross-section view at z = 0.46 mm for AT04 provides visual proof of this phenomenon (Figure 7).

The porosity width profiles in Figure 6b show a smoother variation with multiple porosity peaks at the sample ends. These multiple peaks were attributed to the perturbation in droplet arrangement at the borders that appeared particularly at joint points between ABS and TPU (Figure 5). In the centre, a typical Gaussian distribution profile was evidenced along the width of the specimens.

These Gaussian profiles suggest the presence of porosity gradients across the width for all specimens. The ranking of samples in terms of porosity content promotes AT03 as the best printing condition (overall porosity content, *p* = 0.9 %) and AT02 as the worst one (*p* = 4.7%), according to the statistics in Table 1.

The visual rendering of these width profiles can be read from the cross-section views in Figure 8.

In this figure, the cross-section views in the Y–Z plane are depicted at various positions along the width of the specimens for all printing conditions. The alignment of porosities in the Y-direction and their significant connectivity is symptomatic of difficulties in properly filling the space with droplets during the building up of the prints. This alignment is, in fact, normal to the laying-down direction (Z-direction).

The last profiles depicted in Figure 6c are those corresponding to the porosity evaluation in the X–Z plane, or along the length direction. For the configurations where the interface orientation was normal to the length of the specimens (AT01 and AT02), the effect of the interface on the generated porosity could be easily captured. The presence of a large peak in porosity content at mid-length was more effective for AT02 compared to AT01. For the configuration with an inclined interface (AT03, AT04), there was a monotonous decreasing trend in porosity content because the position of the interface layer moved continuously in the Y-direction (Figure 8). So, the porosity generated at the interface contributed along a large number of longitudinal slices. Thus, there was no sharp variation in porosity content across the sample length in comparison to the former cases (AT01 and AT02). This fact is additionally highlighted by the cross-section views in Figure 9.

The porosity arrangement within the X–Z plane looked similar to the result for the Y–Z plane (Figure 8). This is because the direction of laying down was contained in both cross-section views and the porosities exhibited their highest connectivity in the X- and Y-directions.

The analysis of the porosity content within the ABS and TPU phases revealed an imbalance within the composite. Indeed, the depth profiles in Figure 10 show a change in the magnitude in porosity content within each phase, depending on the printing conditions.

The statistics on porosity content within each phase in Table 1 support this statement. For the intertwining value of the same number, it was found that larger porosity contents in both phases were caused by inclined interfaces (Table 1). This means that inclined interfaces triggered more discontinuities in the composite because of the abrupt change in the droplet trajectories. In a similar way, it was also found that a large number of intertwining droplets added more porosity within each phase when the interface orientation was fixed.

These results suggest a more heterogeneous porosity distribution within each phase, representing, on average, a scatter of 33% if analysed as the difference in porosity content between the phases over the average magnitude. To capture the effect of the interface on the perturbation in phase arrangement, the gradients in porosity contents were explored. For the two cases with horizontal interface (AT01, AT02), the reading of the change in porosity across the interface was easily captured from the porosity depth profiles in Figure 6c. For the remaining conditions (AT03 and AT04), a rotation by 30° in the X–Y plane was needed to analyse the magnitude and extent of the porosity gradient across the interface. Figure 11a depicts the gradient in porosity content across the interface as a function of the printing orientation and number of intertwining droplets at the interface.

The gradients were obtained as the first derivative of the porosity content profiles, and these were shifted at different distance positions for clarity of graphics reading. The largest gradient obtained corresponded to the largest number of intertwining droplets (AT02 and AT04). This result reveals that the laying down of droplets from each side of the interface intended to improve the bonding between ABS and TPU had a side effect of generating a larger porosity gradient at the interface. This result can be explained by the rapid change in toolpath trajectory required to build the interfacial layers.

Figure 11b shows the result of interface tortuosity for all printing conditions. This quantity measures the deviation from a flat interface according to the formula
(8)$$\tau_{k}^{i}=\sum_{j}^{N_{i}}l_{ij}/L_{i}-1\;;\;k=x,\;y,z$$
where $\tau_{k}^{i}$ is the tortuosity at slice i measured in the direction k, $l_{ij}$ is the interface segment j in slice i, and $L_{i}$ is the distance between the end point of the interface at slice i. Using this expression, the tortuosity of a straight interface equals zero. The tortuosity profiles in Figure 11b correspond to $\tau_{Z}^{i}$.

The average tortuosity magnitudes in the X-, Y-, and Z-directions can be expressed as
(9)$$\tau_{k}=\sum_{i}^{N_{k}}\tau_{k}^{i}/N_{k}\;$$
where $N_{k}$ is the number of slices in the direction of measurement k.

Figure 11b demonstrates that there was a minor deviation from the flat interface for all conditions ($\tau_{i}$ < 0.5) throughout the depth of the specimens. However, the tortuosities in the remaining directions were significantly different, as shown in Table 1. The average tortuosity levels were at least one order of magnitude larger than the tortuosity in the Z-direction. The cross-section views in Figure 7, Figure 8 and Figure 9 provide a better understanding of the relevance of this difference in tortuosity levels. If the tilt of the samples during X-ray acquisition is excluded, the difference in tortuosity is first explained by the interface orientation itself as a major factor, and second by the number of intertwining droplets as a minor factor (Table 1).

### 4.2. Overall Tensile Behaviour

The experimental tensile responses of the studied ABS/TPU configurations in Figure 3 suggest a low performance of the printed composites with respect to the properties of the intrinsic materials. For instance, the Young’s moduli for ABS and TPU of 2300 MPa and 527 Mpa, respectively, are well above the slopes in Figure 3c. In addition, the tensile strengths for ABS and TPU were in the range of 42–46 MPa and 10–21 MPa, respectively, which are also above the reported values for the printed composites (Figure 3c). The first reason is that the properties provided by the suppliers are for neat materials obtained by injection moulding. In addition, testing of these materials is performed under displacement rates higher than the ones considered in this study. Additionally, as discussed earlier, 3D printing introduces defects because of the material discontinuities within the phase themselves. On top of that, the weak interface bond can contribute to lowering the ranking of the behaviour. In this section, quantification of the interfacial behaviour is discussed in light of the finite element results. These finite element results, related to the macroscopic model of the ABS/TPU composite, are illustrated in Figure 12.

The computations corresponding to the configurations with horizontal interface (θ = 90°) were used to determine two interfacial parameters (σ*_I_*, δ*_n_*). For these configurations, it was assumed that the tangential separation at the interface plays a minor role when the direction of tensile loading is perpendicular to the interface. The stress component counterplots σ*_yy_* illustrated in Figure 12a corresponded to different levels of load transfer at the ABS/TPU interface. The difference in load transfer was correlated with different magnitudes of the predicted elastic modulus of the composite. Preliminary analysis of the predicted modulus showed that the range of variation of the interface parameters that mostly covered the experimental elastic moduli for the conditions AT01 and AT02 was as follows: (1–30) MPa for σ*_I_* and (0.1–2) mm for δ*_n_*. In fact, the identification of the interface properties (σ*_I_*, δ*_n_*) could only be achieved from AT1 and AT2, as these conditions corresponded to the interface normal to the loading direction. Therefore, the normal separation could not be deduced from AT3 and AT4, as these also depended on the tangential separation. Based on this analysis, a large number (more than 120) of finite element computations was attempted to predict the correlation between the elastic modulus and the interface parameters within the ranges of variations of (σ*_I_*, δ*_n_*) introduced earlier. The other purpose of these computations was to identify the set of interface parameters that fit the experimental results for AT01 (22 ± 0.4 MPa) and AT02 (23 ± 0.2 MPa). The associated counterplot of the predicted elastic modulus is shown in Figure 12b. The counterplot is a representation of the fitting result obtained from the FE results and reflects the effect of the interface parameters on the elastic modulus of the composite. The elastic modulus positively evolved as a function of the ratio σ*_I_*/δ*_n_*. However, it seems that this correlation was not linear. The fitting routine performed on the computation results shows that such correlation can be represented by the function (R² = 0.90):(10)$$E\left(\mathrm{MPa}\right)=8.08+14.81\times \left(\sigma_{I}\left(\mathrm{MPa}\right)-1\right)+\left(8.29/\delta_{n}\left(\mathrm{mm}\right)\right)-\left(0.87/\delta_{n}\left(\mathrm{mm}\right)\right)²$$

Replacing *E* with the experimental value for AT01 or AT02 does not lead to a unique combination of interfacial parameters. The analysis of all combinations leading to an error of less than 1% between the predicted and experimental moduli shows that a correlation exists between interfacial parameters in the form of
(11)$$\delta_{n}\left(\mathrm{mm}\right)=0.26+0.007\times exp\left(3.23\times \sigma_{I}\left(\mathrm{MPa}\right)\right)\;\left|R²=0.99,\;E\left(\mathrm{MPa}\right)=22\;\mathrm{MPa}\;\right.$$
and
(12)$$\delta_{n}\left(\mathrm{mm}\right)=0.28+0.006\times exp\left(3.25\times \sigma_{I}\left(\mathrm{MPa}\right)\right)\;\left|R²=0.99,\;E\left(\mathrm{MPa}\right)=23\;\mathrm{MPa}\;\right.$$

The former correlations represent degeneracy situations that occur when trying to identify the interfacial properties from a macroscopic evaluation of the elastic modulus. In order to unveil the true magnitudes of the interface parameters, local strain measurements were explored.

Figure 13a shows the evolution of the longitudinal strain distribution as a function of the applied load derived from digital image correlation for the case of AT01. It has to be mentioned that the strain value obtained from DIC analysis is a local measurement of the strain field, whereas e is the overall engineering strain obtained as a ratio of the displacement over the initial length. Thus, ε*_yy_* can achieve higher values compared to e, for instance, in the regions where there is a strain localization. In Figure 13a, the strain values near the clamping edges were higher than elsewhere.

The large contrast in stiffness between the phases was the cause of the large stretching of TPU on the bottom and the limited amount of deformation of ABS on the top. In addition, a clear transition in the longitudinal strain was pointed out at the interface, covering a length of few millimetres. The strain heterogeneity within each phase could not be related to the in-plane droplet arrangement. However, evidence of localisation was found more at the lateral faces, especially for the load level e = 2.3%. Figure 13b compares the experimental and numerical longitudinal strain profiles across the interface for different combinations of interface parameters. The computations were performed for an engineering strain of 1%. The longitudinal strain profiles were plotted for x = 0 mm. The evolution of the strain along the Y-axis in Figure 13b appears to have been cut. This was in fact a jump in the strain level due to the contrast between the elasticity properties of ABS and TPU and the marked discontinuity at the interface. The jump was as rapid as a step function because of the small interface thickness. In addition, the strain levels were not uniform across the transverse section. This was due to the local heterogeneity of the material that was triggered by the way the droplets were assembled. In addition, for inclined interfaces (AT03 and AT04), the elongation at both lateral sides of the specimens were not the same because the displacement of the ABS and TPU segments at the lateral edges were not equivalent. In order to increase the strain level within TPU phase (y > 0 mm), the ratio $\sigma_{I}/\delta_{n}$ should be increased. The fitness of the longitudinal strain within the TPU phase came with a larger scatter for the strain within the ABS phase. In fact, due the relatively low strain levels in ABS, the DIC measurements were expected to be noisy. It was thus decided to adjust the interface parameters according to the strain levels in TPU. The best combination that fit the experimental profile was $\sigma_{I}=10.4\;\mathrm{MPa}$, $\delta_{n}=1.35\;\mathrm{mm}$. Similarly, the interface parameters for two intertwining droplets were determined using the same identification scheme. The identification of the tangential component of the interfacial separation was attempted using the cases AT03 and AT04, corresponding to an inclined interface with respect to the loading direction. In these cases, an interfacial shearing took place, which allowed of the maximum tangential separation to be determined.

Figure 14a shows the evolution of the longitudinal strain field measured using DIC for the remaining printing conditions: AT02, AT03, and AT04. In addition to the strain level differences between ABS and TPU, there was a marked strain heterogeneity depicted for AT03 and AT04 corresponding to the effect of the interface orientation. This strain localisation was a preliminary sign for the cracking occurring in both samples at the right size of each specimen. The longitudinal strain profiles along the length of each specimen were used to derive the interfacial properties.

Figure 14b–d show the results based on fixed maximum normal separation ($\delta_{n}$) and maximum normal traction ($\sigma_{I}$) and varied tangential separation ($\delta_{t}$). The comparison between the experimental and numerical strain profiles suggests that the identified shear separation was larger than the normal component of the interfacial separation. Table 2 summarises the identified magnitudes of the interfacial parameters for all printing conditions as well as the overall parameters associated with the overall tensile response of the printed samples, namely, Young’s modulus $E_{Y}$, elongation at bread $\varepsilon_{f}$, and fracture stress $\sigma_{T}$. The comparison between the intrinsic properties of the phases and the properties of the printed samples showed a significant drop in the stiffness performance. The reading of the results in Table 2 suggests that this drop in performance of the printed ABS/TPU was triggered by the interfacial behaviour, which caused the lack of adhesion between ABS and TPU.

The best performing conditions in terms of elongation at break and fracture stress were AT2 and AT4. These corresponded to ABS/TPU interfaces with two intertwining droplets. These conditions also corresponded to the largest maximum traction at the interface and the lowest normal component of the displacement jump at the interface, meaning that these conditions refer to the largest load transfer across the interface.

### 4.3. Microstructural Behaviour

In order to better understand the correlation between the microstructural arrangement and the mechanical response, 3D image finite element-based computations were considered. With the help of massive computation resources, the nodal field results and predicted overall behaviour of printed ABS/TPU for typical elasticity calculations were discussed.

Figure 15 depicts the effect of the resolution on the predicted longitudinal strain distribution and Young’s modulus for all simulated conditions. By binning the 3D TPU/ABS microstructures, the voxel size increased and the resolution was lowered.

In terms of finite element model size, a voxel size between 16 and 218 µm resulted in a model size from 153 million dof down to 0.27 million dof. Within this range, it was found that element sizes lower than 109 µm were not suitable to achieve accurate longitudinal strain fields (Figure 15a). Beyond this limit, only minor differences existed, allowing for the strain field to be more stable against mesh refinement. In addition, the error in the determination of the Young’s modulus remained acceptable < ±5% for all meshes (Figure 15b). However, the stress heterogeneity depicted from low-resolution models was rather poor in detail. This can significantly affect the nature and extent of the damage growth, and in turn, reveals most of its drawback for predicting mechanical properties that are dependent on strain localisation, such as tensile strength or elongation at break.

The fine level of accuracy for both the displacement component UY and the stress intensity distributions were achieved based on an FE model requiring 253.93 million degrees of freedom (Figure 16). In particular, the displacement field coped with the experimental testing conditions. The computation was performed with weak interface conditions, where the stiffness of the interphase layer represented only 1% of the TPU stiffness and about 0.4% of the ABS stiffness. The displacement gradient in the loading direction was altered by the displacement jump at the ABS/TPU interface. The stress intensity distribution exhibited high levels at the clamped ends of the specimen and close to the interphase layer, indicating possible failure sites when the loading was increased.

At the interphase layer, high levels of strain were coherent with the stress heterogeneity taking place because of the presence of stress concentrators, namely, the process-induced porosity. Figure 17 compares the stress intensity distributions for the specimens AT01, AT02, and AT03.

These distributions clearly show the effect of the droplet raster on the development of stress heterogeneity. They also highlight the interfacial area as a region of large stresses. Finally, it seems that the interface orientation played an important role in the nature of the stress localisation. For instance, the specimens AT03 and AT04 exhibited asymmetrical stress profiles along the inclined interface, which suggests that interfacial cracking was likely to have been triggered from only one lateral position (right for AT03 and left for AT04).

Figure 18 shows the predicted longitudinal strain distribution for all printing conditions.

Large strain levels within the TPU phase were common to all conditions. The fine level of microstructural details allowed the strain heterogeneity to be captured around the porosity generated by the processing. When compared to the experimental strain fields derived from DIC, there seems to have been a superior rendering of the computations. However, both the DIC and FE approaches highlight the same strain localisation along the interface occurring for the cases of inclined interfaces AT03 and AT04 (Figure 14a and Figure 18).

Figure 19 shows the finite element results related to the damage-based model for all printing conditions. The damage initiation and growth were decided based on the stress intensity criterion.

As shown in Figure 19a, the predicted stress intensity counterplots highlight the interfacial region as the one where the stress intensity criterion was first verified. The subsequent evolution of the stress distribution showed lower levels at these regions. This evolution compares fairly with the DIC results shown in Figure 13 and Figure 14. There was, however, a difference in the damage growth kinetics attributed to the effect of the interface orientation (Figure 19b). Although the three stages of typical damage kinetics were predicted, namely, initiation, growth, and saturation, the damage level prior to rupture seems to have been more significant for AT01 and AT02 than the other samples. This is attributed to the damage localisation, which can be observed experimentally in Figure 14 and Figure 15 for inclined interfaces. Appendix A are provided to better highlight the damage kinetics for two samples (Appendix A).

## 5. Conclusions

This study concludes that the 3D printing of an ABS/TPU composite using a droplet-based additive manufacturing route resulted in a large porous network connectivity and an acceptable amount of porosity (<5%). In addition to the necking observed between successive droplets, three types of porosity were associated with the material discontinuities generated during the printing process: space between the successive lines of droplets, and at the interface and at the inner side of the external frame. These porosities were correlated with the abrupt change in the toolpath trajectory. Another conclusion about the microstructural arrangement concerns the material anisotropy evidenced from the porosity content profiles. This anisotropy was enhanced by the differences in porosity content and arrangements within the ABS and TPU phases.

This study also concludes that the interfacial behaviour was found to be the limiting factor for the mechanical performance based on the qualification of the interface bonds. In fact, the interfacial behaviour may have resulted in stronger bond than the phase properties depending on the compatibility between the phases. In the present case, the ABS and TPU phases did not lead to this type of strong bond. This conclusion is supported by the digital image correlation results and the finite element predictions. The interfacial properties were found to be sensitive to the quality of the interface. For instance, the maximum interfacial traction was found to be twice as large when the number of intertwining droplets was doubled. Finite element computations are able to capture the effect of droplet arrangement and demonstrate the relevance of microstructural-based finite element models. The predictions of the damage kinetics highlight the role of the interface orientation with respect to the loading direction and allow for a better understanding of the strain localisation evolution. As a prospect, there is room for improving the quality of the bond in ABS/TPU, as shown by the trend demonstrated by the number of intertwining droplets. A way to improve the quality of the bond is to increase the number of intertwining droplets to achieve a more progressive interface, which is possible through the AM process considered in this study. The influence of a progressive interface will be conducted in a future work.

## Figures and Tables

**Figure 1 polymers-14-01013-f001:**
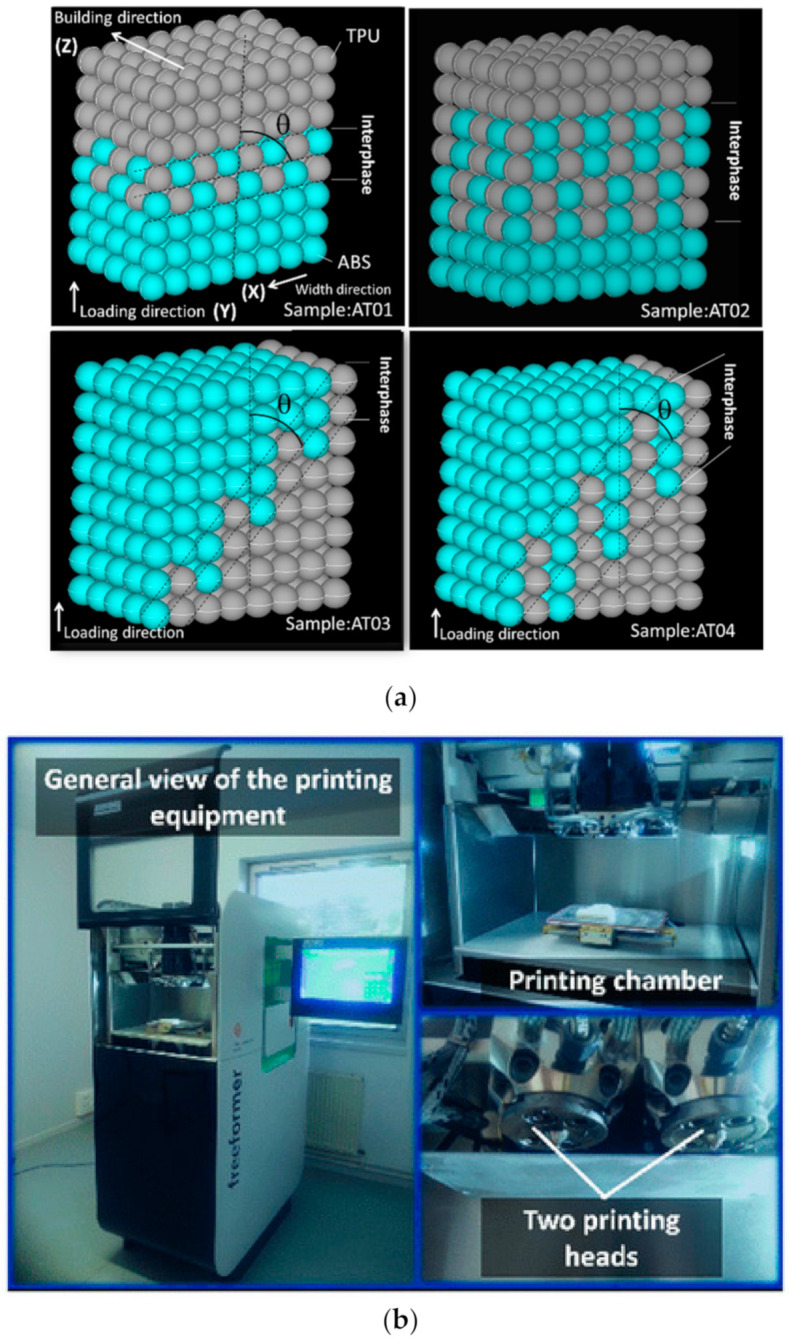
Printing configurations in droplet-based additive manufacturing of a polymer composite. (**a**) Illustration of the printing configurations AT01: one intertwining droplet, θ = 90°; AT02: two intertwining droplets, θ = 90°; AT03: one intertwining droplet, θ = 60°; AT04: two intertwining droplets, θ = 60°. (**b**) Snapshot of the droplet-based additive manufacturing process.

**Figure 2 polymers-14-01013-f002:**
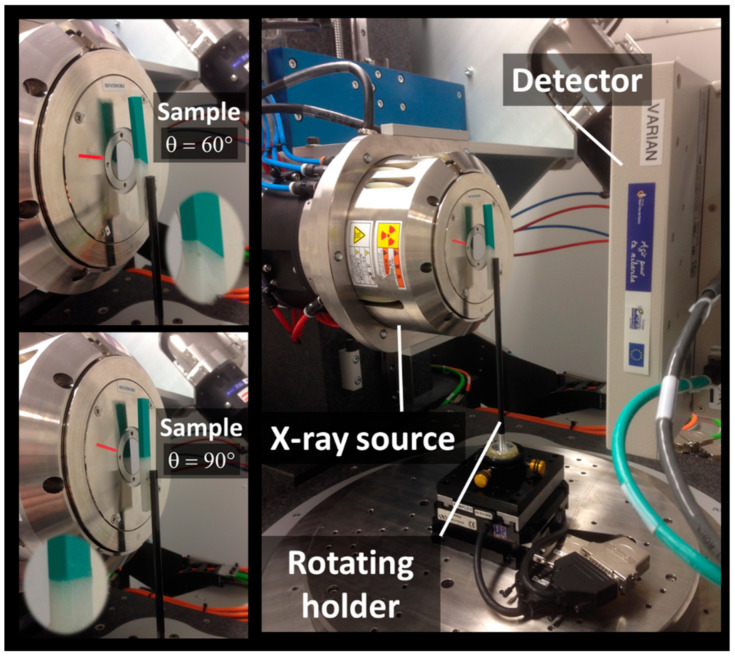
X-ray micro-tomography setup showing the positioning of the printed ABS/TPU samples in front of the X-ray source.

**Figure 3 polymers-14-01013-f003:**
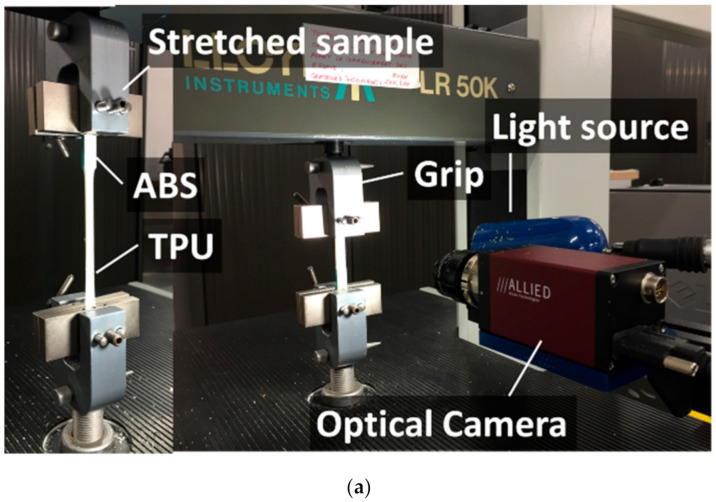
Experimental testing setup: (**a**) Overview of the setup with digital image correlation equipment; (**b**) views of ruptured samples exhibiting a layered structure; (**c**) stress–strain curves derived from the force–displacement response for all studied conditions.

**Figure 4 polymers-14-01013-f004:**
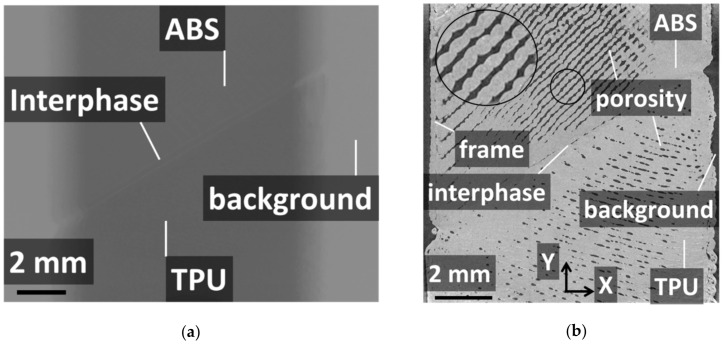
X-ray micro-tomography results: (**a**) Radiographic image of sample AT03 showing the relative contrast between the external air (background), TPU, ABS phases, porosity, and interphase region; (**b**) in-plane cross-section view showing the porous network and the interface layer between ABS and TPU.

**Figure 5 polymers-14-01013-f005:**
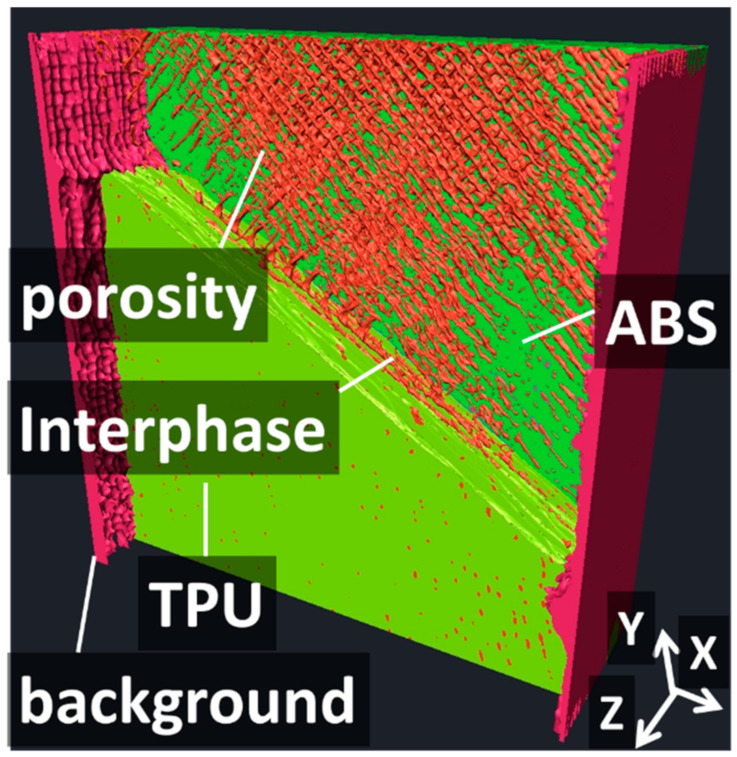
Perspective view showing the layering structure, porosity network, and roughness in the ABS-TPU composite, revealed using X-ray micro-tomography.

**Figure 6 polymers-14-01013-f006:**
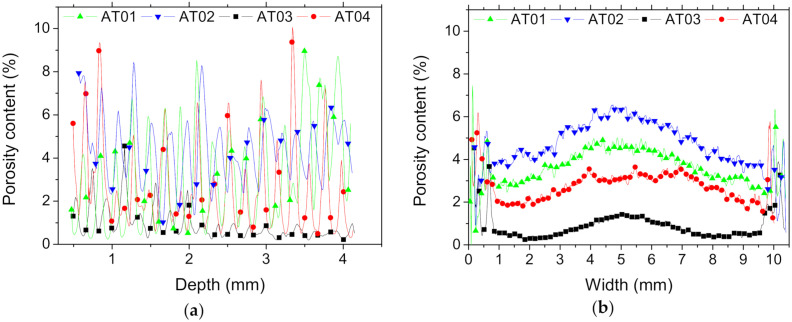
Porosity profiles in: (**a**) depth Z, (**b**) width, and (**c**) along the largest dimension (Y) as a function of the interface orientation and the number of intertwining droplets.

**Figure 7 polymers-14-01013-f007:**
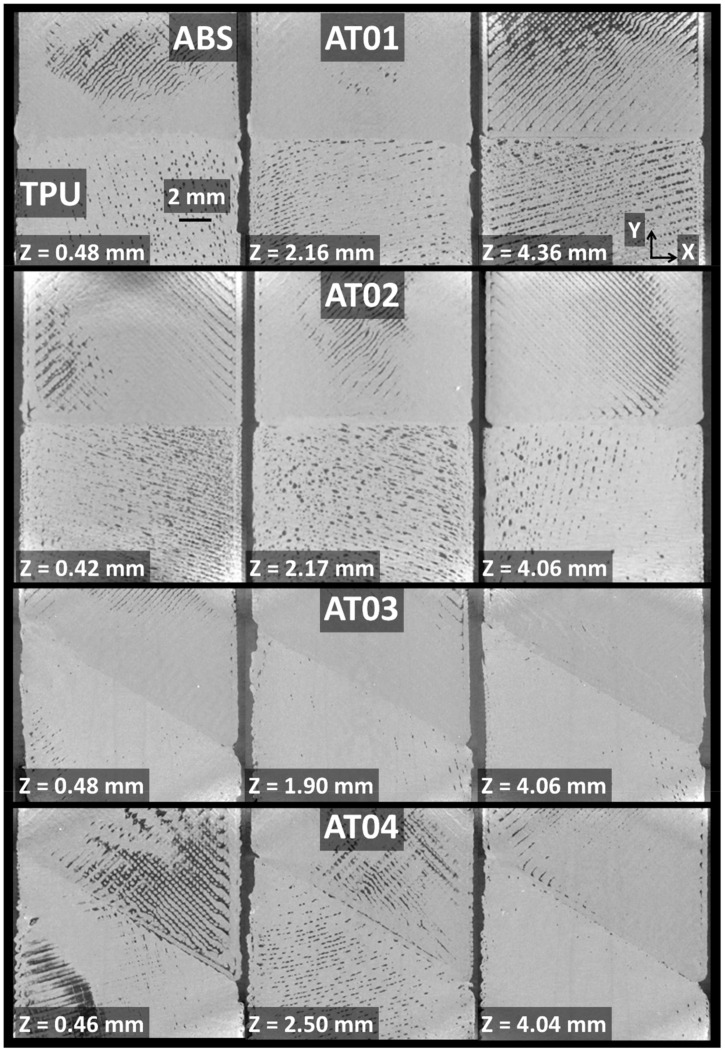
X–Y in-plane cross-section views (dimensions 10 mm × 10 mm) showing the porosity arrangement at different depth positions (Z) for all printed ABS/TPU specimens.

**Figure 8 polymers-14-01013-f008:**
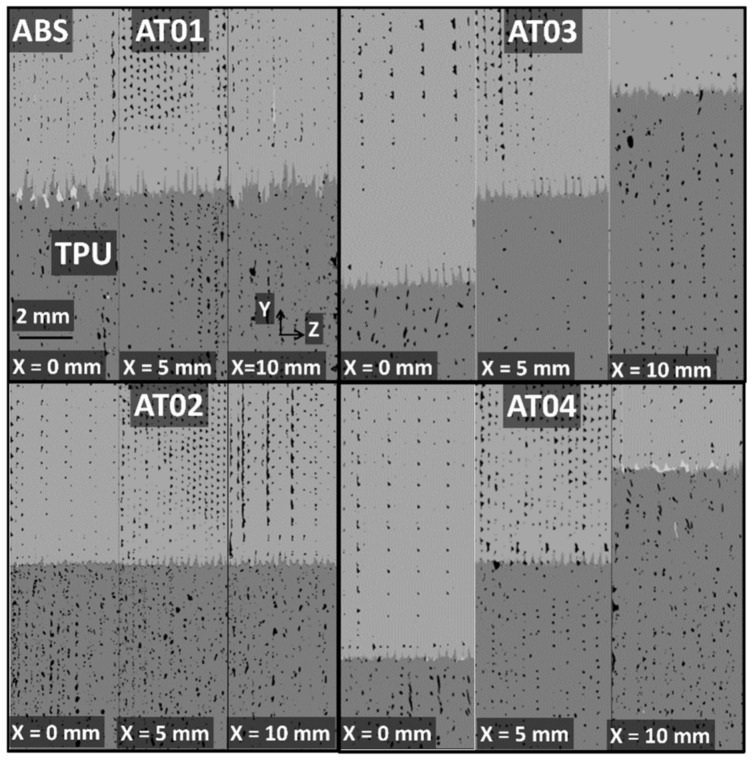
Y–Z cross-section views (dimensions 10 mm × 4 mm) at different width positions (X) for all printed ABS/TPU samples. (The phase contrast was achieved after image processing.)

**Figure 9 polymers-14-01013-f009:**
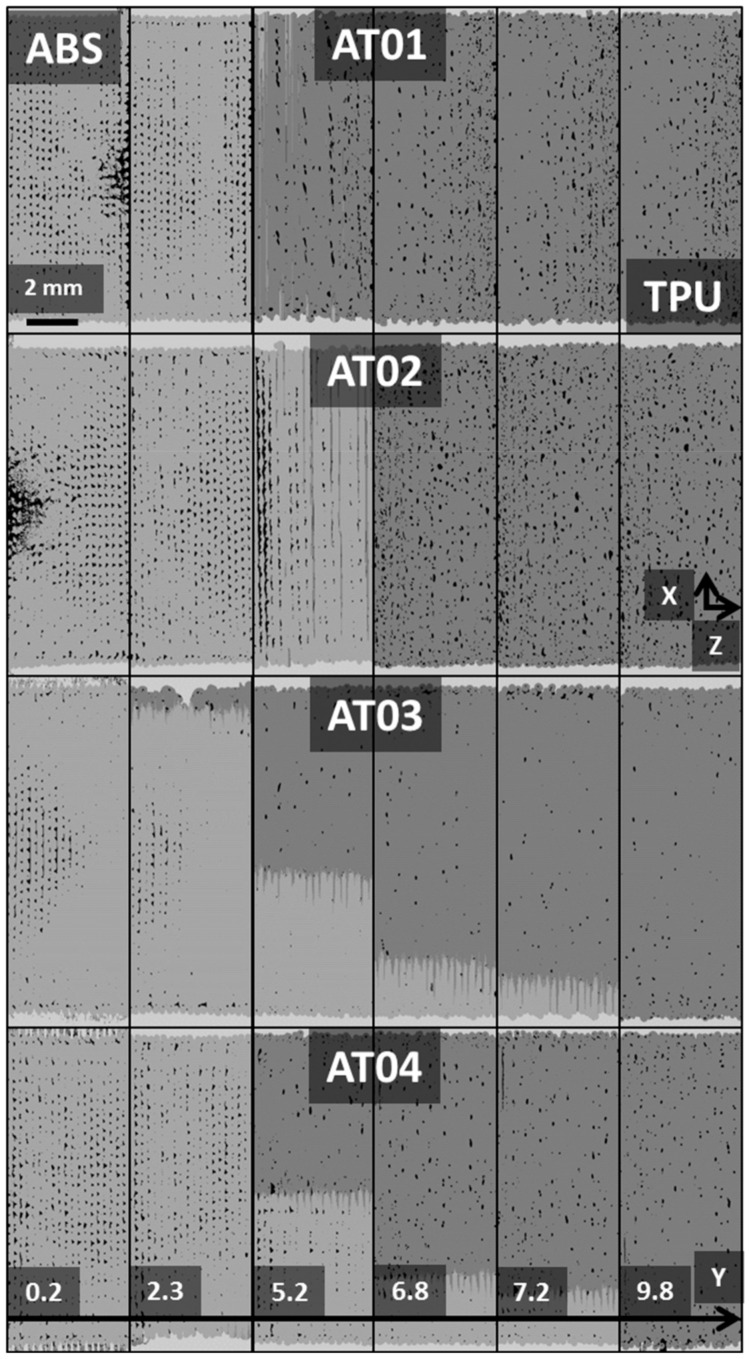
X–Z cross-section views (dimensions 10 mm × 4 mm) at different length positions (Y) for all printed ABS/TPU samples. (The phase contrast was achieved after image processing.)

**Figure 10 polymers-14-01013-f010:**
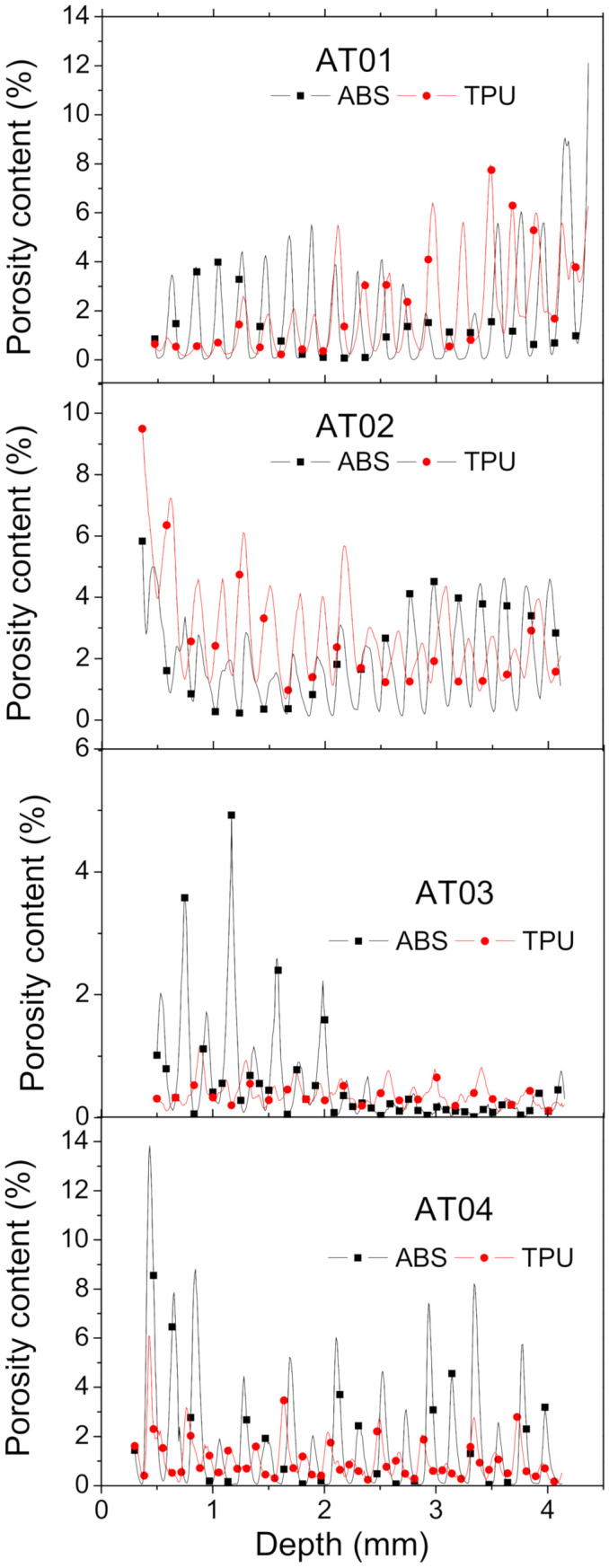
Depth porosity profiles in each phase of the ABS/TPU composite for all printing conditions.

**Figure 11 polymers-14-01013-f011:**
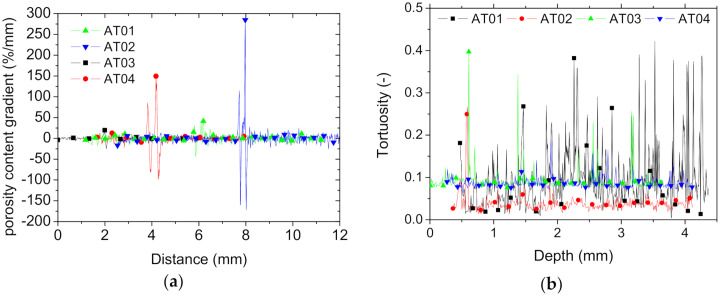
Results of porosity analysis: (**a**) gradient of porosity content across the interface and (**b**) interfacial tortuosity through the depth of the printed ABS/TPU composite for all printing conditions.

**Figure 12 polymers-14-01013-f012:**
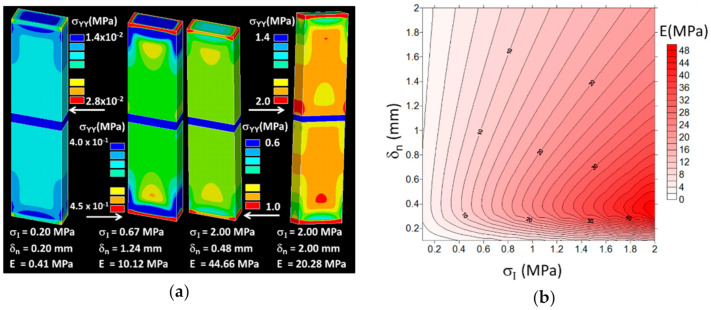
Finite element results showing (**a**) the predicted stress component (σ*_yy_*) distribution for a horizontal TPU/ABS interface (σ*_I_*, δ*_n_*) and selected combinations of interface parameters (σ*_I_*, δ*_n_*) and (**b**) the predicted elastic modulus E counterplot of the ABS/TPU composite based on 120 finite element computations with different combinations of the interface parameters σ*_I_*, δ*_n_*.

**Figure 13 polymers-14-01013-f013:**
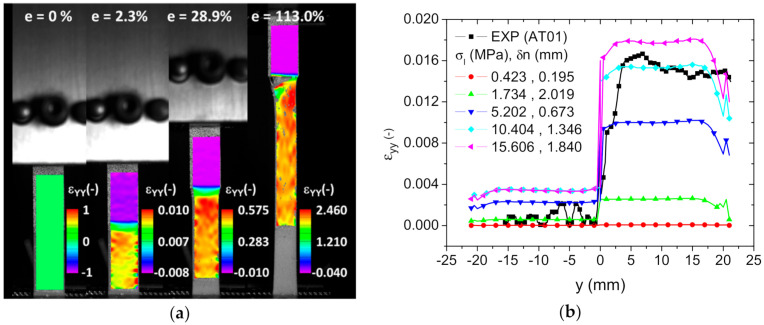
DIC-based results: (**a**) longitudinal strain component ε*_yy_* as a function of the load level e (height extension in percent) for AT01; (**b**) identification of the interface parameters (σ*_I_*, δ*_n_*) from the strain profile across the interface using DIC measurements.

**Figure 14 polymers-14-01013-f014:**
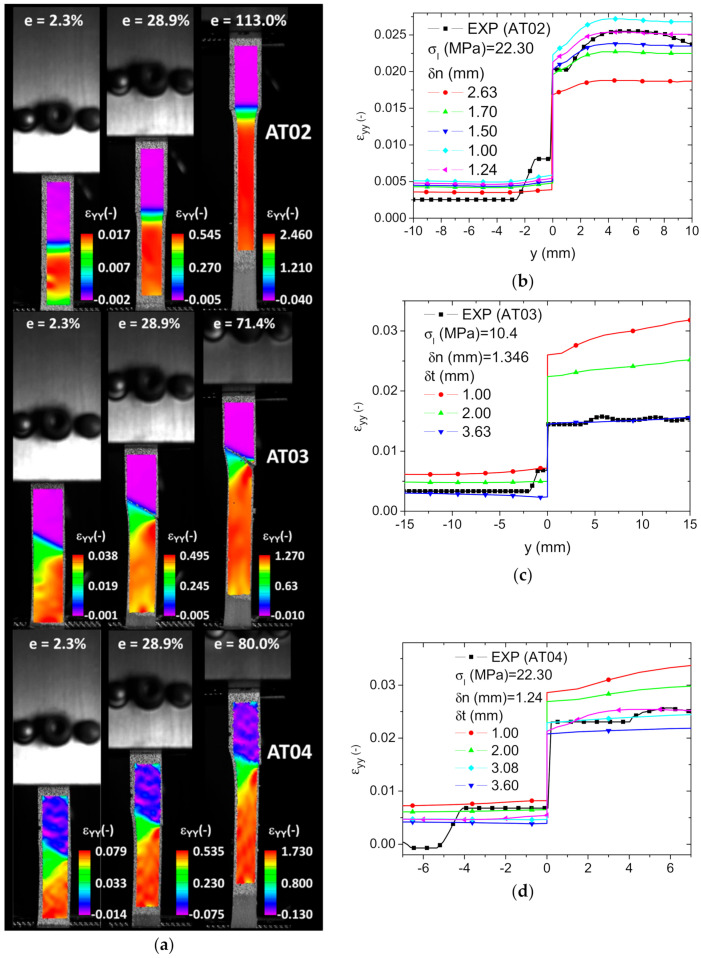
DIC and finite element results compared: (**a**) longitudinal strain component ε_yy_ as a function of the load level e (height extension in percent) for AT02, AT03, and AT04; (**b**–**d**) comparison of the finite element and DIC longitudinal strain profiles and identification of the tangential component of the ABS/TPU interfacial separation.

**Figure 15 polymers-14-01013-f015:**
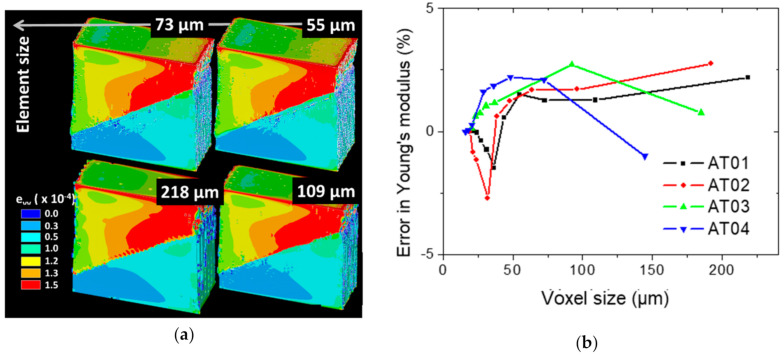
Finite element results: (**a**) longitudinal strain distribution within the AT03 specimen as a function of the element size; (**b**) effect of resolution (element size = voxel size) on the prediction of the Young’s modulus for all printing conditions.

**Figure 16 polymers-14-01013-f016:**
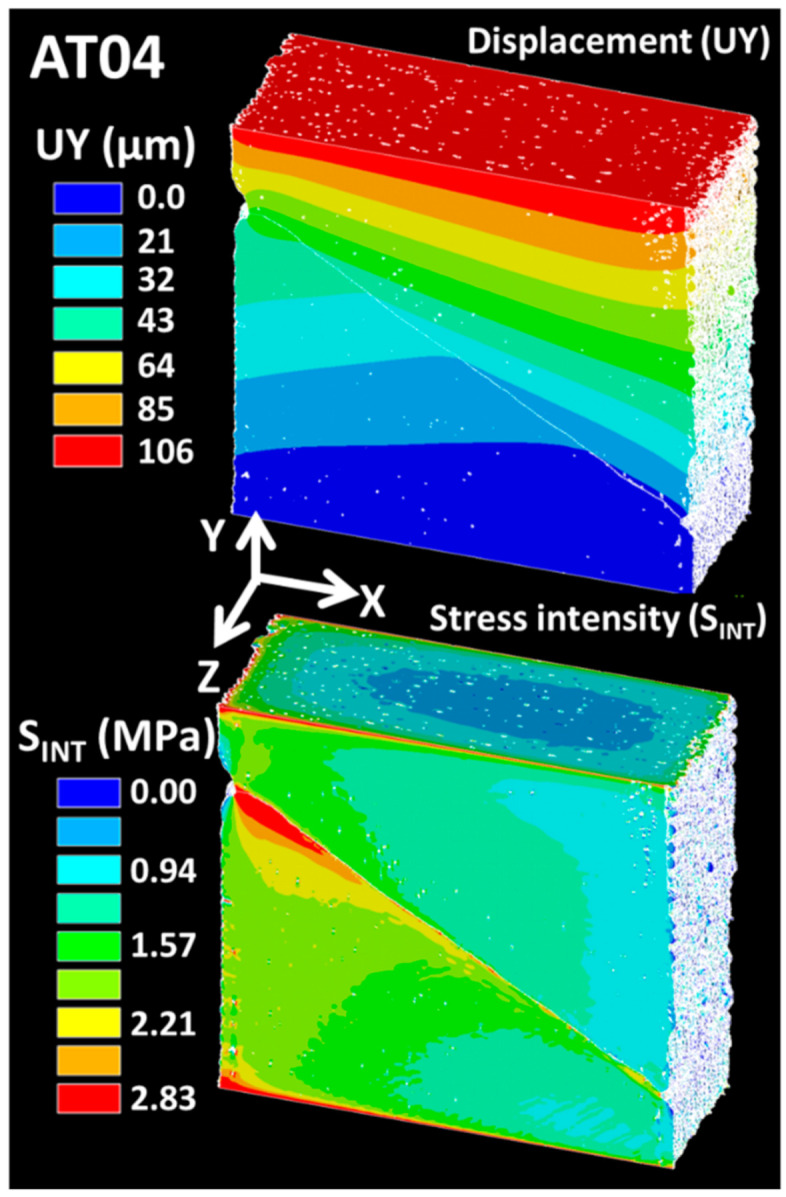
Displacement component UY and stress intensity fields issued from 3D image-based finite element computation for AT04.

**Figure 17 polymers-14-01013-f017:**
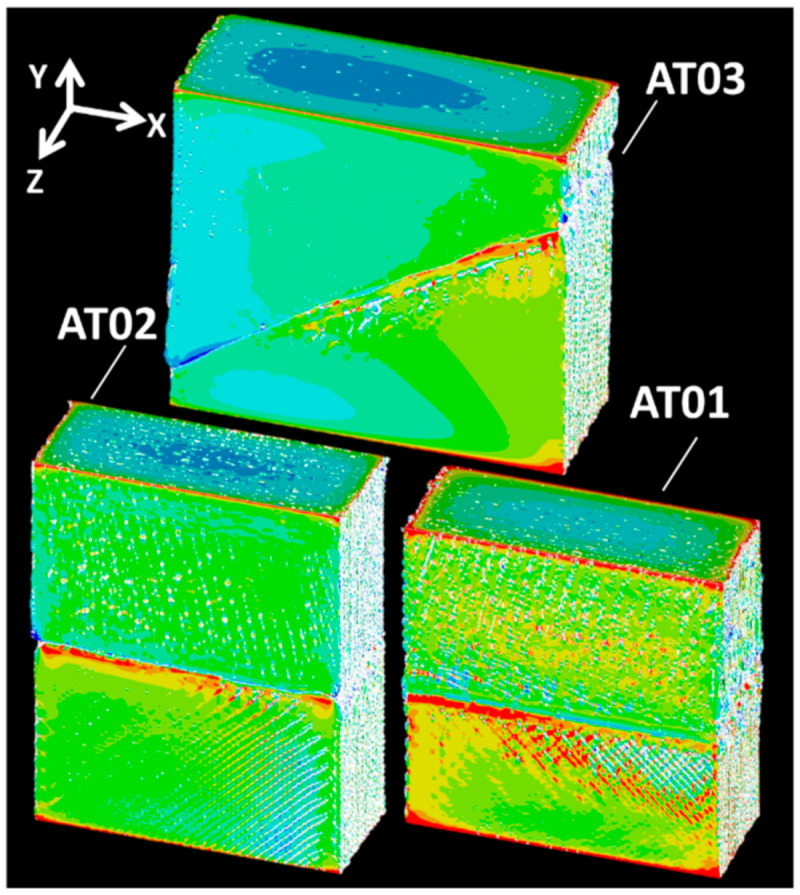
Displacement component UY and stress intensity fields issued from 3D image-based finite element computation for AT04.

**Figure 18 polymers-14-01013-f018:**
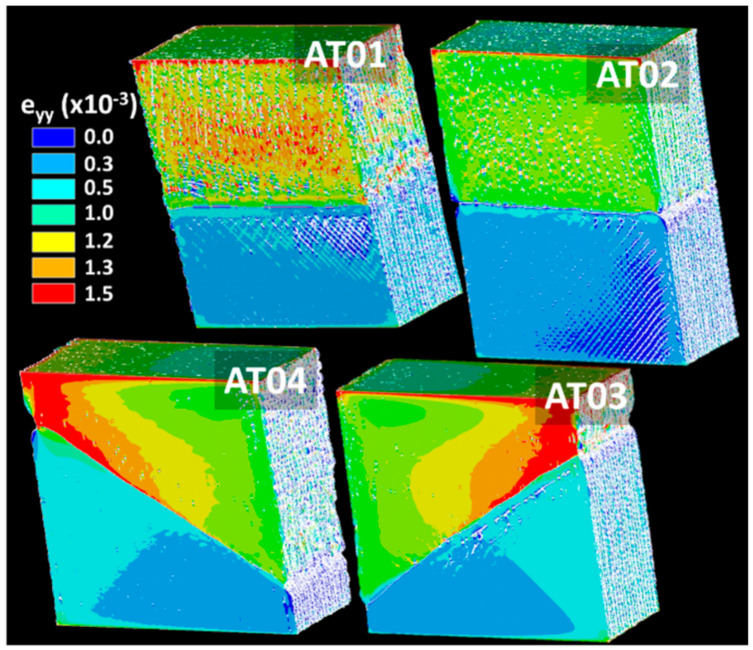
Longitudinal strain component as a function of the printing orientation and number of intertwining droplets.

**Figure 19 polymers-14-01013-f019:**
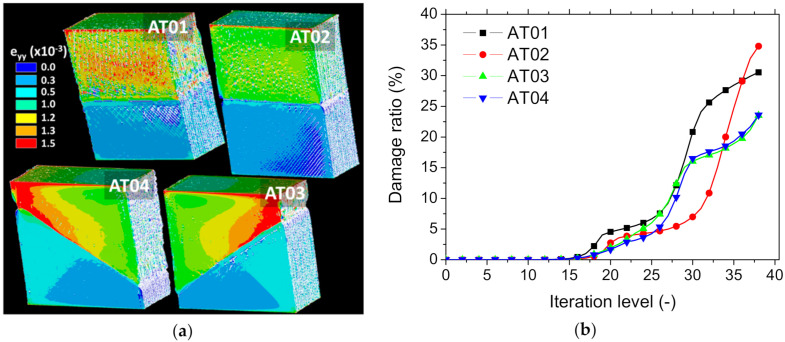
Finite element results: (**a**) stress intensity evolution as a function of the printing conditions and the level of damage; (**b**) damage ratio evolution for all printing conditions.

**Table 1 polymers-14-01013-t001:** Microstructural attributes derived from image analysis of X-ray micro-tomography images.

Printing Condition			Porosity Content (%)			Interfacial Tortuosity (-)		
#	n (-)	θ (°)	Total	ABS	TPU	τ*_X_*	τ*_Y_*	τ*_Z_*
AT01	1	90	3.61	2.11	1.50	10.1	1.9	0.094
AT02	2	90	4.69	2.97	1.72	5.5	2.2	0.039
AT03	1	60	0.89	0.43	0.46	9.0	22.0	0.090
AT04	2	60	2.75	1.12	1.63	7.3	34.2	0.087

**Table 2 polymers-14-01013-t002:** Identification results of the interfacial behaviour as a function of the printing conditions.

Printing Condition			Interfacial Parameters			Tensile Behaviour		
#	n (-)	θ (°)	σ*_I_* (MPa)	δ*_n_* (mm)	δ*_t_* (mm)	$E_{Y}\;(\mathrm{MPa})$	ε*_f_* (%)	σ*_T_* (MPa)
AT01	1	90	10.40	1.35	-	29.40	115	5.46
AT02	2	90	22.30	1.24	-	25.94	201	8.33
AT03	1	60	10.40	1.35	3.63	31.99	68	4.75
AT04	2	60	22.30	1.24	3.08	33.73	141	6.23

## Data Availability

The data presented in this study are available on request from the corresponding author.

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
