# Peer review of "Interfacial Behaviour in Polymer Composites Processed Using Droplet-Based Additive Manufacturing"

_polymers, 2022, doi:10.3390/polym14051013_

Round 1

Reviewer 1 Report

The manuscript investigates the effect of the resolution of 3D-printing using droplet-based additive manufacturing technology.
A composite of ABS and TPU polymers tested prepared by four different configurations and several printing resolution. The resultant elements focusing on the interface created between the two polymers 
have been checked with various techniques. Porosity and several parameters describing the interface (Young modulus, etc.) and the strains have been determined. The conclusions are adequate. 

A linear scaling is found between the maximum of interfacial traction and the number of intertwining droplets but the explanation is missing. It should be added in the revised manuscript. 

The text is well written, I found one typo: page 3line 92: Both horizontal and inclined interfaces ... (omission of the indefinite article ). The figures, however, have to be improved. 
Figure 1: the subfigures should have the same size and symmetrically positioned. 
Figure 2: unnecessarily big and there is a slight difference between the two pictures (theta)
Figure 7: horizontal lines separating the several cases would help the reader.
Figure 14: figures should be rearranged: left column is fig a, right column. figs b,c,d
Figure 15: figure arrangement is awkward, at least the pictures should be centralized
Figure 19: symbols of 19 a and b should be added onto the figure

All of the tables are numberes as table 1; table numbers should be checked

Based on the above I recommend the manuscript for publication after revision.

Author Response

Comments and Suggestions for Authors

The manuscript investigates the effect of the resolution of 3D-printing using droplet-based additive manufacturing technology.
A composite of ABS and TPU polymers tested prepared by four different configurations and several printing resolution. The resultant elements focusing on the interface created between the two polymers have been checked with various techniques. Porosity and several parameters describing the interface (Young modulus, etc.) and the strains have been determined. The conclusions are adequate. 

A linear scaling is found between the maximum of interfacial traction and the number of intertwining droplets but the explanation is missing. It should be added in the revised manuscript. 

The text is well written, I found one typo: page 3line 92: Both horizontal and inclined interfaces ... (omission of the indefinite article ).

The sentence is corrected.

The figures, however, have to be improved. 

Sure please check the new layout of the figures according to the reviewer comment.

Figure 1: the subfigures should have the same size and symmetrically positioned. 

The modification requested is done. The new figures are now symmetrically positioned with the exact same size.

Figure 2: unnecessarily big and there is a slight difference between the two pictures (theta)

The size of the figure is adjusted and the figure adapted to show more features of the experimental setup.

Figure 7: horizontal lines separating the several cases would help the reader.

The figure is modified as requested, now horizontal lines separate the four cases.

Figure 14: figures should be rearranged: left column is fig a, right column. figs b,c,d

The figure is arranged according to the comment

Figure 15: figure arrangement is awkward, at least the pictures should be centralized

We are sorry for the layout, we tried our best with this new version. The figure size is adjusted accordingly.

Figure 19: symbols of 19 a and b should be added onto the figure

The symbols a and b are already present on the left side. Now they are centered like with the previous figures. In addition, we modified the layout to a two-column representation.

All of the tables are numberes as table 1; table numbers should be checked

We are sorry for such a mistake, Table 2 is now corrected and citation checked.

Based on the above I recommend the manuscript for publication after revision.

We do appreciate the opinion of the reviewer. We are also ready to take further comments out of the first revision.

Reviewer 2 Report

This article deals with Interfacial behaviour in polymer composites processed using 2 droplet-based additive manufacturing.

The results are supported by the digital image correlation and the finite element predictions. Was it compared to the experimental results? What about the percentage of deviation?

Please give the snap shot of droplet-based additive manufacturing experimental set up.

Please mention the exact application of the present study.

Most of the references are clubbed together such as 6-8 and 9-13. Please try to discuss the pros and cons of the individual article.

Author Response

Reviewer 2 :

Comments and Suggestions for Authors

This article deals with Interfacial behaviour in polymer composites processed using 2 droplet-based additive manufacturing.

The results are supported by the digital image correlation and the finite element predictions. Was it compared to the experimental results? What about the percentage of deviation?

Digital image correlation is based on experimental data, it is used as a full-field contactless technique of the strain field. These experimental strain fields are compared to the numerical results. The reviewer points out a good remark about the accuracy. In the case of the digital image correlation, the deviation is related to the resolution of the measurements, which is for DIC about 10 microstrain. In the case of the FE computation, this depends on the solver. In the present case, we used a preconditioned for which the error tolerance is adjusted to 10-8.

For both cases, the accuracy of the measurement is quite high which ensures a high confidence in the reported results.

Amendment in section 2 (experimental layout) “The accuracy of the measurement is as small as 10 microstrain, which is well below the achieved strain levels upon loading. ”  3 (modelling) + section 3 “The PCG error tolerance is adjusted to 10-8, which is the recommended value for elasticity problems. ”.

In addition, the numbering of the section is checked and updated.

 Please give the snap shot of droplet-based additive manufacturing experimental set up.

The following figure is added to Figure 1 to illustrate the AM process.

Please mention the exact application of the present study.

ABS material is usually used for outdoor applications due to the excellent, chock resistance, dimensional stability,  and electric insulations. It is for instance widely used in consumer products, communication equipment, automotive parts, home appliance.  TPU on the other hand is known for its elasticity, resistance to abrasion, and has common use as ABS in automotive industry, appliance, construction, and recreation industry. The combination of the tough and elasticity is needed in some application like in automotive industry to allow demping while maintaining a suffient mechanical resistance. Thus, this study focuses on understanding how the composite made of ABS and TPU can be used without degrading the performance of the two phases by looking at the interface bounding when using a bi-material parts in the context of a droplet 3D printing technique. Indeed, 3D printing technologies are quite attractive in general permit accessing quite complicated structures with the ability to combine different materials. Thus understating the mechanical response is quite important as its helps design structures with desired mechanical strength.

Amendment in introduction section : “The materials considered for this study are Acrylonitrile Butadiene Styrene (ABS) and Thermoplastic polyurethane (TPU). ABS material is usually used for outdoor applications due to the excellent, chock resistance, dimensional stability, and electric insulations. It is for instance widely used in consumer products, communication equipment, automotive parts, home appliance. TPU on the other hand is known for its elasticity, resistance to abrasion, and has common use as ABS in automotive industry, appliance, construction, and recreation industry. The combination of the tough and elasticity properties in a single material is needed in some applications like in automotive industry to allow damping under cycling loading while maintaining a sufficient mechanical resistance. Thus, this study focuses on understanding how the composite made of ABS and TPU can be used without degrading the performance of the two phases. This can be done by looking at the interface bounding when using a bi-material parts in the context of a droplet 3D printing technique. Indeed, 3D printing technologies allows more degrees of freedom to adapt the quality of the interface as in the droplet-based technology to achieve adaptable designs with the ability to combine different materials. This approach is considered in this study to demonstrate the leverage on the design that can help improve the bonding between ABS and TPU.

Most of the references are clubbed together such as 6-8 and 9-13. Please try to discuss the pros and cons of the individual article.

The references are now discussed in more details. Please check the new version.  Most of the inputs were doe in the introduction section :

“The manufacturing of technical parts from a digital model is an effective way to produce complex geometries [1, 2]. As reported by [1], AM is developed to conceptualise models for products but quickly to process materials. Zai et al. [2] also reports that AM is an advanced manufacturing process for the fabrication of 3D functional components with a limited dependence to tooling and fixture. Despite the large amount of time needed to manufacture a single part by Additive Manufacturing (AM) technologies compared to traditional processes such as injection moulding or extrusion[3], AM is still an attractive way [3-5]. Indeed, the fabrication cycle for AM requires only a CAD model instead of preparing, adapting or producing a new tool. As described by Baumers et al.[3] AM has in that sense a huge economic impact on the manufacturing sector because of the reduced tooling costs. According to Berman [5], this reduction of cost can be also attributed to automation that allows AM be autonomous not requiring constant attention from the operators during the production cycle. The economic impact has been coving a large number of industries but as reported in the review paper of Mousa et al. [4] key sectors such as aerospace, automotive, medical industries and electronics are sharing most of the AM focus. In terms of fabrication cycle, once the CAD model is prepared, series of tool paths are the only intermediate step to transform the virtual sketch into a real part [6-8]. As explained in the paper of Pandey[6], the series of tool paths requires a slicing of the CAD model into series of thin layers. The thickness of these layers determines the balance between the surface finish and the building time. According to Hague et al. [7], the outcome of the slicing procedure results in critical factors such as surface finish, resolution, accuracy that engineers need to constantly adapt to allow AM to be a viable manufacturing technology. This is why early contributions such as the one on stereolithography by Jamison et al.[8] tackled the deviation errors in design tessellation, which is the basic procedure for converting the 3D design into a set of triangular patches.

Waiving the obligation for tooling means also increasing the offer for personalised products that AM is able to fulfil[5]. There is no doubt that several industries understood clearly this potential by directing a substantial investment towards AM-based products [9-13]. The review paper by Sing et al. [9] shows that the leading market for AM-based products is the consumer products and electronics with up to 21% of the total market shares. Automatic and medical sectors come in the second and third positions, respectively. In the automotive industry, for instance, the market needs to lighter, safer, and cost effective products with as much as possible short lead time are the main value drivers for AM according to Giffi et al. [10] prospect paper. In the aerospace sector, safety concerns with regards to reliability of 3D printed parts is still a challenge but some leading projects to use AM processes to manufacture parts have been granted substantial funding such as to produce air-cooling ducts, plastic interior parts, leading-edge blades, even entire spacecraft structures [11]. In the medical sector, Peltola et al.[12] point out the advantage of AM overall conventional route for biocompatible part for customising implants or engineered body parts from medical scans of patients. A huge segment of market is dedicated to AM technologies such as tissue engineering for bone and cartilage repair applications. For civil engineering applications, Krimi et al. [13] demonstrate that design flexibility versus the production rate is still the challenging balance for AM to compete with standard routes in construction such as prefabrication or casting on site. The growing interest on AM technologies comes with some aspirations to solve the problems of part inaccuracy and finishing state [14-16], to improve the productivity [9], and to gain in performance [17, 18]. For instance, Zhou et al. [14] applied surface response and ANOVA analysis to optimise the process parameters in stereolithography technology to achieve optimal dimensional accuracy and build time. Most of the performance limitations come from mechanical anisotropy and the loss in strength and stiffness driven by the presence of particular arrangement of defects. For instance, Dawoud et al.[18] compared the rendering of FDM (Fused Deposition Modelling) with injection moulding and showed that ABS specimens manufactured using injection moulding exhibited the highest mechanical performance. Hague et al.[7] compared the mechanical performance of specimens printed using streolithography and laser sintering process under different building orientations. The study concluded on different trends with respect to the processing route. While streolithography based specimens exhibited isotropic properties, laser sintering process generated more anisotropic parts. Puebla et al. [17] showed however that stereolithography may generate anisotropic behaviour especially if the part orientation is combined with different aging trends. Huang et al.[15] studied the shape shrinkage responsible for design dimension inaccuracy and developed a methodology for anticipating geometric errors. Depending on the complexity of the designs, geometrical inaccuracy can be also related to the presence of support material needed for printing overhangs. Thus, attempts have been made to decrease the influence of support material as suggested by Paul et al. [16] by using algorithmic approach to minimise the flatness and cylindricity errors. 

There is, generally, no balance between AM technologies with regards to their sensitivity to process-induced defects. For instance, Fused Deposition Modelling (FDM) is known to produce two-dimensional discontinuities because the process is based on the laying down of 1D continuous features, i.e., the filaments. In the 3D space, this obviously generates a particular arrangement of three-dimensional porosities. As a result, the part contains a well-ordered porosity network evidenced by several imaging techniques such as X-ray micro-tomography [19]. The origin of mechanical anisotropy, and especially the lack of performance in the building direction results from this process-induced porosity. In the case of stereolithography, the laser drawing leaves no material discontinuity within the plane, and the liquid state of the resin used in such technology ensures continuity of the material both in the plane of deposition and along the building direction [20]. This means that stereolithography has a better mechanical rendering compared to FDM but has some limitations especially to produce parts with closed porosities [20]. In addition, stereolithography requires the feedstock material to be photo-sensitive and generates more operating costs. These factors narrow down the spectrum of applications for this type of technology.

In this study, the effect of process-induced defects in droplet-based additive manufacturing is investigated with respect to the tensile performance of a polymeric composite. This technology naturally produces 3D discontinuities related to the successive laying down of individual droplets. In particular, this study combines 3D imaging tool based on X-ray micro-tomography, finite element computation and testing experiments to reveal the nature, magnitude and extent of the process-generated defects as well as the interfacial effect on the mechanical response. The materials considered for this study are Acrylonitrile Butadiene Styrene (ABS) and Thermoplastic polyurethane (TPU). ABS material is usually used for outdoor applications due to the excellent, chock resistance, dimensional stability, and electric insulations. It is for instance widely used in consumer products, communication equipment, automotive parts, home appliance. TPU on the other hand is known for its elasticity, resistance to abrasion, and has common use as ABS in automotive industry, appliance, construction, and recreation industry. The combination of the tough and elasticity properties in a single material is needed in some applications like in automotive industry to allow damping under cycling loading while maintaining a sufficient mechanical resistance. Thus, this study focuses on understanding how the composite made of ABS and TPU can be used without degrading the performance of the two phases. This can be done by looking at the interface bounding when using a bi-material parts in the context of a droplet 3D printing technique. Indeed, 3D printing technologies allows more degrees of freedom to adapt the quality of the interface as in the droplet-based technology to achieve adaptable designs with the ability to combine different materials. This approach is considered in this study to demonstrate the leverage on the design that can help improve the bonding between ABS and TPU.

Reviewer 3 Report

This paper studies the mechanical behavior of the interface between two different polymers, namely in the paper ABS and TPU. The authors study four kind of interfaces, with two different orientations and one and two intertwining droplets.

General comments

The authors find a significant drop in the stiffness performance of the composite in comparison to the stiffness of the two components. This is a logical result because the interface is very narrow, just with one or two interwining droplets. So, why not to study a more progressive interface, with a larger number of droplets? And also: what is the motivation of this study? In the introduction it is explained that this technology “naturally produces 3D discontinuities related to the successive laying down of individual droplets”, but this is not the studied discontinuity. To my understanding, the discontinuity that the paper studies is induced by two different materials. Which is the interest of studying the interface between two different materials and what do the authors expect? The authors should clarify this point that, in my opinion, is very important to motivate the work.

In general, Figures are of poor quality. Comments regarding some figures are given below  

Experimental layout

I think that it is better to use the same perspective for the four samples of Figure 1. I propose to plot figures 2 and 3 in similar sizes. Figure 2 is too large in comparison to Figure 3

Microstructural characteristics

Where is Figure 5(b)? In Figure 6, the right part is missed. The sizes of figures 7, 8 and 9 are quite different.

How are the total values of porosity in Table 1 obtained?

Overall tensile behavior

The authors combine experimental and numerical (FEM) results. They experimentally measure the Young Modulus E and from it, adjust numerically the three material parameters that define the interface, modeled with cohesive elements. They obtain in the experiments: 22 MPa (AT01), 23 MPA (AT02). I cannot find the values for the other two interfaces. After the fitting, the numerical elastic modulus is between 25.94 and 33.73 MPa (It should be Table 2 instead of Table 1). Very much lower than the E of ABS (2300 MPa) and TPU (527 MPa) or an equivalent E of both materials. My question is: how the E value is computed in the experiments? The Young modulus is the slope of the stress strain relationship in a uniaxial test in the elastic regime. Which are the computed magnitudes? The imposed displacement versus the force? In addition, the DIC results are far from being linear, there are very large deformations (for example, e=113%) Which is the relation between e and εyy? I would expect similar values and an equivalent Young modulus similar to the one of the softer material. I think that the authors must show the curve used to compute the Young modulus in the experimental test. This point is important because the experimental E (22 MPa) is used to fit the interfacial parameters obtaining a numerical E (25.94 MPa) as close as possible to the experimental one. I imagine that both materials are modelled as linear elastic.

The evolution of strain along y-axis in figures 13 and 14 are cut. I suppose the strain is uniform across the transversal direction in AT01 and AT02, but not in AT03 and AT04. In the last two specimens, where is located the evolution of strain? At x=0? At which value of e are these plots?

The values of the fracture stress of the individual materials would be indicated to compare with the values of the Table

The values given for the interfacial parameters given in lines 371 and 371 do not correspond to those of Figure 12(b)

Conclusions

The authors said that this study concludes that the interfacial behavior is found to be the limiting factor for the mechanical performance. This conclusion is obvious.

The authors must do an effort to explain which the interest of the work is, and which conclusions are obtained from it

Author Response

Reviewer 3 :

General comments

The authors find a significant drop in the stiffness performance of the composite in comparison to the stiffness of the two components. This is a logical result because the interface is very narrow, just with one or two interwining droplets.

In Composite engineering, the quality of the bond is not necessarily related to the thickness of the interface. Some composites exhibit strong interface properties because of the nature of chemical bond generated at the interface. We were thus not expecting that the ABS/TPU interface is weak as the compatibility between ABS and TPU was not known prior performing the mechanical tests.

Amendment in introduction section: “Composites exhibit strong or weak interface properties depending on the nature of chemical bond generated at the interface. In the present study both the thickness and orientation of this interface are varied to quantity the quality of ABS/TPU interface properties on the composite tensile behaviour.”

So, why not to study a more progressive interface, with a larger number of droplets?

This is a good idea, we thank the reviewer for pointing out this possibility. We did not look at a higher number of droplets at the interface as the investigation looked at the combination of interface thickness and orientation. In the new version, we introduced a new prospect based on the former comment. Due to the extensive work that has been done in the characterization and testing, we believe that this extension requires more space and will be tackled in another paper.

Amendment in conclusion section “As a prospect, there is a room for improving the quality of the bond in ABS/TPU as shown by the trend demonstrated by the number of the intertwining droplets. A way to improve the quality of the bond is to increase the number of intertwining droplets to achieve a more progressive interface, which is possible through the AM process considered in this study. The influence of a progressive interface will be conducted in a future work.”

And also: what is the motivation of this study? In the introduction it is explained that this technology “naturally produces 3D discontinuities related to the successive laying down of individual droplets”, but this is not the studied discontinuity. To my understanding, the discontinuity that the paper studies is induced by two different materials. Which is the interest of studying the interface between two different materials and what do the authors expect? The authors should clarify this point that, in my opinion, is very important to motivate the work.

We do not fully agree with this viewpoint, the combination of the two materials is done through two different nozzles (See new Figure 1). This means that in one part of the specimen, the discontinuity is created only by one material not the combination of the two. We do however agree that close to the interface both material will influence the discontinuity in the composite.

Amendment in introduction section: “In a printed composite structure, the material discontinuity may exhibit a complex trend that this study intends to quantity. For instance, in a two-phase composite defects created in each phase are not the same as these depend on the thermal behavior of the intrinsic phases during the laying down. But close to the interface both phases are expected to contribute to the defects created by material discontinuity.”

In general, Figures are of poor quality. Comments regarding some figures are given below

We already addressed the figure layout from comments of the first reviewer. Please find above the details of the modifications on the figures.  

Experimental layout

I think that it is better to use the same perspective for the four samples of Figure 1.

The figure was modified according to reviewer 1 remark. In addition, we modified the perspective view of case ST02 as requested to be in line with the perspective view of the other cases.

 I propose to plot figures 2 and 3 in similar sizes. Figure 2 is too large in comparison to Figure 3

Figure 2 was modified according to reviewer 1 remark. In addition, we adjusted the size of figure 3 according to reviewer 2 remark. Please check the new layout.

Where is Figure 5(b)? In Figure 6, the right part is missed.

Sorry there is no Figure 5b, this is a copy paste mistake from Figure 4. Modification in the figure citation is done in the new version.

There are only three subfigures attached to Figure 6. There is no right part missing.

Amendment new figure title: “Figure 5. Perspective view showing the layering structure, porosity network and roughness in ABS-TPU composite revealed using X-ray micro-tomography.”

The sizes of figures 7, 8 and 9 are quite different.

We could not avoid that because the parallepipedic shape of the acquired samples. As mentioned in section 2, the size of the acquired samples are  10 mm × 10 mm × 4 -5 mm. This produces cross-section views different because of the reduced z-dimension. The dimensions are now mentioned in all figure legends.

Please check the new figure titles :

Figure 7. XY in-plane cross-section views (dimensions 10 mm x 10mm) showing the porosity arrangement at different depth positions (Z) for all printed ABS/TPU specimens.

Figure 8. Y-Z cross-section views (dimensions 10 mm x 4mm) at different width positions (X) for all printed ABS/TPU samples (The phase contrast is achieved after image processing).

Figure 9. X-Z cross-section views (dimensions 10 mm x 4 mm) at different length positions (Y) for all printed ABS/TPU samples (The phase contrast is achieved after image processing).

How are the total values of porosity in Table 1 obtained?

The porosity levels are measured from image processing using the following formula :

                                                                             (1)

This formula is added in the new version.

Amendment in section 2:

Porosity content is derived from segmented images, where porosity phase appear in binary images as black voxels () and white voxels () are assigned to either ABS or TPU phase. Further processing treatment is done to avoid counting the surrounding external area and also to distinguish between the porosity levels in each phase.

                                                                             (1)

Where  is the grey level of voxel  from the domain , where porosity level is measured.

In addition, renumbering of all formulas is also done.

Overall tensile behavior

The authors combine experimental and numerical (FEM) results. They experimentally measure the Young Modulus E and from it, adjust numerically the three material parameters that define the interface, modeled with cohesive elements. They obtain in the experiments: 22 MPa (AT01), 23 MPA (AT02). I cannot find the values for the other two interfaces.

The identification of the interface properties (sI, dn) can only be achieved from AT1 and AT2 as these conditions correspond to the interface normal to the loading direction. Thus, the normal separation cannot be deduced from AT3 and AT4 as these also depend on the tangential separation.

Amendment in results section: “In fact, the identification of the interface properties (sI, dn) can only be achieved from AT1 and AT2 conditions as these conditions correspond to the interface normal to the loading direction. Therefore, the normal separation cannot be deduced from AT3 and AT4 as these also depend on the tangential separation.”

After the fitting, the numerical elastic modulus is between 25.94 and 33.73 MPa (It should be Table 2 instead of Table 1). Very much lower than the E of ABS (2300 MPa) and TPU (527 MPa) or an equivalent E of both materials.

Table 2 title is corrected according to the former comment.

My question is: how the E value is computed in the experiments?

Young’s modulus is measured from the initial slope of the stress-strain curves. Please check the raw data as force (N) – displacement (micron)

We added in the manuscript the mechanical response for all materials

Regarding the low values of Young’s moduli, please note that the values given for ABS and TPU are the values related to injection moulding processing not 3D printing. These values are obtained under displacement rates higher than the ones considered in this study. 3D printing would induce defects (porosity) that also lower the values on intrinsic materials. On top of that the weak interface bond measured contributes to lower the ranking of the behaviour.

Amendment in section 2: “From the tensile experiments, both engineering strain and stress are derived and Young’s modulus is measured from the initial slope of the stress-strain curves.”

Figure 3 updated

(c)

addition to Figure 3 title

Figure 3. Experimental testing setup: (a) Overview of the setup with digital image correlation equipment; (b) views of ruptured samples exhibiting a layered structure; (c) stress – strain curves derived from the force – displacement response for all studied conditions.

+ in section 4: “The experimental tensile responses of the studied ABS/TPU configurations in Figure 3 suggests a low performance of the printed composites with respect to the properties of the intrinsic materials (for instance, Young’s moduli for ABS and TPU of 2300 MPa and 527 MPa are well above the slopes in Figure 3c). The first reason would be that the properties provided by the suppliers are for neat materials obtained by injection moulding. In addition, testing of these materials are performed under displacement rates higher than the ones considered in this study. Also, as discussed earlier, 3D printing introduces defects because of the material discontinuities within the phase themselves. On top of that the weak interface bond can contribute to lower the ranking of the behaviour. In this section, quantification of the interfacial behaviour is discussed at the light of the finite element results.  ”

The Young modulus is the slope of the stress strain relationship in a uniaxial test in the elastic regime. Which are the computed magnitudes? The imposed displacement versus the force?

Yes, this is added now in the new version. Please check the answer above.

In addition, the DIC results are far from being linear, there are very large deformations (for example, e=113%) Which is the relation between e and εyy? I would expect similar values and an equivalent Young modulus similar to the one of the softer material.

Εyy is obtained from DIC analysis which means that this quantity is a local measurement of the strain field whereas e is the overall engineering strain obtained as a ratio of the displacement over the initial length. Thus εyy can achieve higher values compared to e for instance at the regions where there is a strain localization. In figure 13 for instance the reviewer can see that the strain values near the clamping edges are higher than elsewhere.

Amendment in section 4 : “It has to be mentioned that the strain value obtained from DIC analysis is a local measurement of the strain field whereas e is the overall engineering strain obtained as a ratio of the displacement over the initial length. Thus, εyy can achieve higher values compared to e for instance at the regions where there is a strain localization. In figure 13a the strain values near the clamping edges are higher than elsewhere.

I think that the authors must show the curve used to compute the Young modulus in the experimental test. This point is important because the experimental E (22 MPa) is used to fit the interfacial parameters obtaining a numerical E (25.94 MPa) as close as possible to the experimental one. I imagine that both materials are modelled as linear elastic.

This is now done in the new version. Please find in Figure 3c the results of the testing. Also, in the review letter we provide the raw measurements in the form of force – displacement curves. 

The evolution of strain along y-axis in figures 13 and 14 are cut. I suppose the strain is uniform across the transversal direction in AT01 and AT02, but not in AT03 and AT04. In the last two specimens, where is located the evolution of strain? At x=0? At which value of e are these plots?

This is not a cut but a shift in the values due to the contrast between the elasticity properties of ABS and TPU. The shift is rapid as a Dirac function because of the small interface thickness.

In addition, the strain is not uniform across the transverse section. This is due at the local heterogeneity of the material that is triggered by the way the droplets are assembled. In addition, the reviewer can notice the situation of inclined interfaces where the elongation at both lateral sides of the specimens cannot be the same simply because the displacement of the ABS and TPU segments are not the same.

The reviewer is right the counterplots are reported for x=0 mm. The engineering strain used is 1%.

Amendment in section 4: “The evolution of strain along y-axis in figures 13b appears to be cut. This is in fact a shift in the strain level due to the contrast between the elasticity properties of ABS and TPU and the marked discontinuity at the interface. The shift is as rapid as a step function because of the small interface thickness.”

+

“In addition, the strain levels are not uniform across the transverse section. This is due at the local heterogeneity of the material that is triggered by the way the droplets are assembled. In addition, for inclined interfaces (AT03 and AT04), the elongation at both lateral sides of the specimens are not the same because the displacement of the ABS and TPU segments at the lateral edges are not equivalent.”

+

“The computations are performed for an engineering strain of 1%. The longitudinal strain profiles are plotted for x = 0 mm.”

The values of the fracture stress of the individual materials would be indicated to compare with the values of the Table

These values were not available from the suppliers, we reported values from the literature crossing various references.

Amendment in section 4:

“In addition, tensile strength for ABS and TPU are in the range of (42 – 46) MPa, and (10 – 21) MPa, respectively, which are also above the reported values for the printed composites (Figure 3c).”

The values given for the interfacial parameters given in lines 371 and 371 do not correspond to those of Figure 12(b)

Please note that the representation in Figure 12b is based on 120 FE computations not on equation (9) which is obtained after fitting these data. In addition, the isocontour scheme requires a limited number of contours to ease the reading of the figure. In the new version, we plotted the function reported in equation 9 to avoid the confusion and we used also smaller bounds for dn and si according to the previous version to reflect the predicted data: “elastic moduli for the conditions AT01 and AT02 is as follows: (1 – 30) MPa for sI and (0.1 – 2) mm for dn.“.

Please check the new rendering in Figure 12b.  + amendment in section 4 : “The counterplot is a representation of the fitting result obtained from the FE results and reflects the effect of the interface parameters on the elastic modulus of the composite.”

Conclusions

The authors said that this study concludes that the interfacial behavior is found to be the limiting factor for the mechanical performance. This conclusion is obvious.

We are sorry to say that this is not obvious. This is related to the compatibility between the two phases. For some combinations, the interface quality is stronger than the intrinsic phase properties. This study provided a quantification of the interface bonding effect since we were able to predict what is the interfacial resistance and the maximum separation length. It also shows the influence of the interface droplet interphase thickness and orientation.

Amandement in conclusion section “This study also concludes that the interfacial behaviour is found to be the limiting factor for the mechanical performance based on the qualitification of the interface bonds. In fact, the interfacial behaivour may result on stronger bond than the phase properties depending on the compatiblity between the phases. In the present case, ABS and TPU phases do not lead to this type of strong bond.”

The authors must do an effort to explain which the interest of the work is, and which conclusions are obtained from it

This study shows a procedure that combines several tools (DIC, FEM, Micro-tomography) to quantify mechanical response of printed bi-materials. The obtained data are important for the prediction of the strength of printed parts used for engineering design. We also provided the motivation behind this study in the introduction section based on the reviewer 1 remark. Please check the new version.

Additional modifications in section 1: “In fact, overmolding process can be considered as the conventional counterpart of AM for the design of composite structures [21]. Although a substantial progress has been made in the area of multi-material injection molding, the complexity of the designs brought by the growing need for light weight and highly peformant composites limits the use of overmolding process to simlpified designs. Another critical issue addressed in many papers related to the overmoulding technique is the quality of the interface [22]where in these conventional process, there is a limited laverage to adjust the interface properties by directly playing on the process deposition. This context allows the AM to be a step ahead for provinding laverage both at the material side by selecting the appropriate material combination and at the process side by adjusting locally the interface properties. ” + added two references related to injection moulding.

Round 2

Reviewer 2 Report

Authors have carried out extensive numerical studies. It should be reflected in the title. 

Authors have carried out revisions satisfactorily.

Reviewer 3 Report

The authors have done an effort to answer correctly all my questions and the paper has improved its quality.  I think that presents interesting results and can be published.

However, there is a point that I do not understand and I want to share with the authors. They have added the following information:

  • The experimental tensile responses of the studied ABS/TPU configurations in Figure 3 suggests a low performance of the printed composites with respect to the properties of the intrinsic materials (for instance, Young’s moduli for ABS and TPU of 2300 MPa and 527 MPa are well above the slopes in Figure 3c). The first reason would be that the properties provided by the suppliers are for neat materials obtained by injection moulding.
  • In addition, tensile strength for ABS and TPU are in the range of (42 – 46) MPa, and (10 – 21) MPa, respectively, which are also above the reported values for the printed composites (Figure 3c)

My impression is that they are observing the behaviour of the softer material (Figure 13), at least in the elastic regime, and the modulus of elasticity in Figure 3c is the one of this material. I would test each material separately, because the mechanical properties of them are not for 3D printing conditions but for injection moulding, so we can not compare the experimental results to the reference ones. I think that the testing machine can arrive to 250 MPa (10KN / 40 mm2), enough to break both materials.